# BOOSTING THE VISUAL INTERPRETABILITY OF CLIP VIA ADVERSARIAL FINE-TUNING

**Shizhan Gong, Haoyu Lei, Qi Dou & Farzan Farnia**
Department of Computer Science and Engineering
The Chinese University of Hong Kong
{szgong22, hylei22, qdou, farnia}@cse.cuhk.edu.hk

## ABSTRACT

CLIP has achieved great success in visual representation learning and is becoming an important plug-in component for many large multi-modal models like LLaVA and DALL-E. However, the lack of interpretability caused by the intricate image encoder architecture and training process restricts its wider use in high-stake decision making applications. In this work, we propose an unsupervised adversarial fine-tuning (AFT) with norm-regularization to enhance the visual interpretability of CLIP. We provide theoretical analysis showing that AFT has implicit regularization that enforces the image encoder to encode the input features sparsely, directing the network's focus towards meaningful features. Evaluations by both feature attribution techniques and network dissection offer convincing evidence that the visual interpretability of CLIP has significant improvements. With AFT, the image encoder prioritizes pertinent input features, and the neuron within the encoder exhibits better alignment with human-understandable concepts. Moreover, these effects are generalizable to out-of-distribution datasets and can be transferred to downstream tasks. Additionally, AFT enhances the visual interpretability of derived large vision-language models that incorporate the pre-trained CLIP an integral component. The code of this paper is available at https://github.com/peterant330/CLIP_AFT.

## 1 INTRODUCTION

Recent advancements in vision-language foundation models have successfully facilitated multi-modal representation learning that aligns heterogeneous inputs into a unified embedding space. A prominent example is CLIP (Radford et al., 2021), which employs contrastive learning on a large dataset of paired text and images to associate images with their textual descriptions. This method effectively positions similar concepts close together and demonstrates high efficacy in downstream tasks such as zero-shot classifications (Saha et al., 2024) and open-vocabulary object detection (Minderer et al., 2024). Additionally, CLIP serves as a built-in component in several large vision-language models including DALL-E (Ramesh et al., 2021) and LLaVA (Liu et al., 2024).

The flexibility and generalizability of CLIP render it highly attractive for high-stake decision-making processes such as medical diagnosis (Kim et al., 2024) and autonomous driving (Xu et al., 2023b). In these safety-critical applications, neural network interpretation becomes paramount. However, several studies (Chefer et al., 2021a; Li et al., 2022; 2023) have demonstrated that CLIP has unsatisfying interpretability. This deficiency can be partly attributed to the inherent nature of ViT (Fel et al., 2022) and partly to the non-smoothness induced by false negatives during the contrastive training (Gao et al., 2021; Wu et al., 2021). To elucidate this, we present the results of two popular explainable AI techniques in Fig. 1, namely saliency maps (Simonyan et al., 2013; Selvaraju et al., 2017) and network dissection (Bau et al., 2017). Both the simple gradient map and Grad-Cam saliency map from the CLIP's image encoder look noisy and random. Additionally, the number of concept detectors within the encoder, a common measure of the model interpretability, is limited. Given that CLIP's image encoder is frequently employed as a plug-in component of various tasks, enhancing its interpretability could significantly improve the reliability of downstream applications.

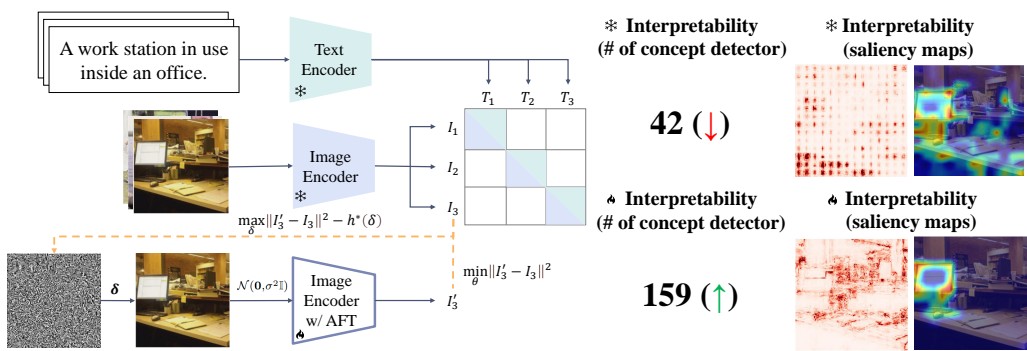

Figure 1: After adversarial fine-tuning (AFT), the image encoder of CLIP can generate more sensible saliency maps and contain more concept detectors, becoming more interpretable.

Adversarial training (AT, Madry et al., 2017), initially designed to bolster the adversarial robustness of models, has been empirically and theoretically demonstrated across numerous studies to also enhance the interpretability of neural networks (Ross & Doshi-Velez, 2018; Xu et al., 2023a). It has been shown to suppress irrelevant features and accentuate the discriminative features in saliency maps (Ross & Doshi-Velez, 2018; Kim et al., 2019; Etmann et al., 2019; Chalasani et al., 2020; Shah et al., 2021; Wang et al., 2021; Gong et al., 2024a), and enable networks with different architectures to encode common causal patterns (Ren et al., 2023). Furthermore, by enforcing smoothness in network predictions within the local neighborhood of data points, AT improves the local linearity of the network, a widely-used metric for model interpretability (Li et al., 2020; Khan et al., 2024). Regarding the application of AT to multi-modal models such as CLIP, the recent FARE method (Schlarmann et al., 2024) utilizes AT to improve the robustness of the CLIP model against norm-bounded perturbations. However, extending the application of AT to boost the interpretability of multi-modal vision-language models remains underexplored and is still limited to uni-modal classification settings in the literature. Additionally, it is unclear whether the AT's effect on interpretabiliy exhibits zero-shot generalizability or can be transferred to downstream tasks.

In this paper, we systematically study the relationship between AT and the visual interpretability of CLIP. We focus on adversarial fine-tuning (AFT) to mitigate the computational demands and optimally utilize the extensive pre-trained models available. We begin with an objective function minimizing the $L_2$-distance between visual-language similarity scores w/wo perturbations. We demonstrate that AFT with properly-designed norm regularization can be reformulated as standard training with dual-norm regularization on the input gradients. Therefore, to promote the sparsity of gradient-based saliency maps, we specifically consider a Huber loss function as the dual norm in the optimization, which leads to a piecewise quadratic penalty function in the proposed AFT min-max optimization. The designed AFT optimization problem concentrates on the interpretability of the fine-tuned model, which differentiates our proposed method from the existing FARE method leveraging AT to improve the robustness of the CLIP model against $\ell_\infty$ norm-bounded adversarial perturbations. Subsequently, we derive an upper bound for the minimization objective function, which is independent of the text embedding. By minimizing this upper bound, we can isolate the text encoder and fine-tune only the image encoder. The fine-tuned image encoder remains compatible with the original text encoder and downstream large language models tuned with CLIP embeddings.

We empirically compare the visual interpretability w/wo AFT. The results demonstrate that AFT significantly enhances the quality of the saliency maps (see Fig. 1). The improvement is independent of the specific saliency map techniques employed, encompassing fundamental feature attribution methods (e.g., Simple Gradient, Grad-Cam) as well as more advanced approaches (e.g., M2IB (Wang et al., 2023), Grad-ECLIP (Zhao et al., 2024)). It is also transferable to out-of-distribution data and applicable to various downstream tasks, even when tuned on relatively small datasets. We also evaluate the interpretability of the image encoder through clip-dissect (Oikarinen & Weng, 2022) and network dissect (Bau et al., 2017). Our findings reveal that AFT enhances the alignment of neural activations with human-understandable concepts and promotes more object-centric activations. Additionally, we demonstrate that AFT improves the visual interpretability of large vision-language models when integrated with CLIP as a plug-in component. Our main contributions are as follows:

- We propose an unsupervised AFT with norm-regularization that can enforce the pre-trained CLIP to encode visual features sparsely and thus improve its visual interpretability.

- We theoretically prove the duality relation between the regularized norms of adversarial perturbations and the input gradients to explain how AFT improves visual interpretability.

- We provide various quantitative and qualitative experimental results to support that AFT improves the visual interpretability of CLIP and its derived large vision-language models.

## 2 RELATED WORK

**Adversarial Robustness of CLIP Model.** While imperceptible perturbations may significantly change the network prediction (Szegedy et al., 2013; Athalye et al., 2018), existing literature have proposed many methods (Madry et al., 2017; Sinha et al., 2017; Zhang et al., 2019; Cohen et al., 2019) to improve the robustness of neural networks towards such adversarial attacks. Studies (Mao et al., 2022; Gu et al., 2024) also find the CLIP model vulnerable to adversarial attacks. To improve the adversarial robustness of the CLIP model, Wang et al. (2024b) applies an efficient adversarial training strategy during the training phase of the CLIP; Mao et al. (2022) puts forward a supervised fine-tuning strategy to improve the robustness of specific downstream tasks; Wang et al. (2024a) retrains a more robust image encoder guided by the original image encoder; Schlarmann et al. (2024) proposes an unsupervised fine-tuning method that achieves robustness with only the image branch being fine-tuned. Deviated from their motivation, our work explores the possibility of building a more interpretable CLIP model via a tailored adversarial fine-tuning strategy.

**Interpretations in Computer Vision.** Interpretability is defined as the capability to provide explanations understandable to humans (Doshi-Velez & Kim, 2017). To understand the decision process of the black-box neural networks, a series of work focuses on feature attribution, where saliency maps are used to highlight the important features (Simonyan et al., 2013; Ribeiro et al., 2016; Lundberg & Lee, 2017; Sundararajan et al., 2017; Selvaraju et al., 2017; Shrikumar et al., 2017; Zhang & Farnia, 2023; Muzellec et al., 2024; Gong et al., 2024b; Ye & Farnia, 2024). Another stream of work give mechanistic interpretation to the neural network by understanding which concept each neuron is associated with (Bau et al., 2017; Oikarinen & Weng, 2022; Kalibhat et al., 2023; Ahn et al., 2024; Bai et al., 2024; Gandelsman et al., 2024). Networks with higher interpretability would have more concept detectors and each neural should align better with a specific concept. We apply both feature attribution and network dissection to evaluate the visual interpretability of CLIP.

**CLIP Interpretability.** There are a few existing works focus on the interpretation of CLIP, with most of them focusing on concept-based explanations which ties to decompose the visual embeddings into multiple concepts represented by text embeddings (Yun et al., 2022; Moayeri et al., 2023; Chen et al., 2023; Gandelsman et al., 2023; Oikarinen et al., 2023; Yang et al., 2023; Chattopadhyay et al., 2024; Bhalla et al., 2024). In terms of visual feature attribution, Li et al. (2022; 2023) modify the architecture of the CLIP to improve the quality of saliency maps; Wang et al. (2023) proposes a CLIP attribution method based on the information bottleneck principle; Gandelsman et al. (2023) decomposes the image representation across image tokens to generate heatmaps; Zhao et al. (2024) presents a gradient-based method for visual explanation. While most of these work applies black-box or gray-box attributions, i.e., using no or only the gradient information of the last layers, they excel at explaining the interaction between visual and text branches with less attention paid to interpreting the entire visual encoder. Our work, instead of proposing a new feature-attribution technique for interpreting the prediction of CLIP, would focus on understanding and improving the inherent interpretability of the visual encoder, which is generalizable and transferable to downstream tasks.

## 3 ADVERSARIAL FINE-TUNING FOR MORE INTERPRETABLE CLIP

In this section, we introduce the AFT algorithm and give a theoretical explanation of why it can improve the interpretability of the image encoder. The method overview is illustrated in Fig.1. We take a pre-trained CLIP model with any backbone and adversarially fine-tune its visual encoder using a relatively small, image-only dataset, guided by the objectives in Eq.7. The framework allows for flexible regularization choices based on the desired properties of the saliency maps. Our findings indicate that this adversarial fine-tuning enhances the quality of the saliency maps produced by CLIP and improves the alignment of its neurons with human-understandable concepts.

## 3.1 NORM-REGULARIZED ADVERSARIAL TRAINING FOR CONCISE INTERPRETATIONS

We first define the form of adversarial fine-tuning objective in the context of language-image contrastive learning. The CLIP is composed of an image encoder and a text encoder, which encodes the language-image pair into their representations. We use $\mathbf{x}$, $I_{\mathbf{x}}$, and $T_{\mathbf{x}}$ to represent the input image, its image embedding, and the paired text embedding, respectively. Assume $I_{\mathbf{x}}$ and $T_{\mathbf{x}}$ have been normalized into unit length, the language-image similarity is the cosine similarity between $I_{\mathbf{x}}$ and $T_{\mathbf{x}}$, i.e., $T_{\mathbf{x}}^T I_{\mathbf{x}}$. AFT aims to improve the robustness of the fine-tuned image encoder $f_\theta(\cdot)$ towards minor perturbations so that the language-image similarity remains the same w/wo perturbations. To this end, we define the objective function as follows:

$$\min_\theta \mathbb{E}_{\mathbf{x} \sim \mathcal{D}_{\text{train}}} \max_{\delta_{\mathbf{x}}} \frac{1}{2}(T_{\mathbf{x}}^T \mathbb{E}_{\mathbf{z} \sim \mathcal{N}(\mathbf{0}, \sigma^2 \mathbb{I})}[f_\theta(\mathbf{x} + \mathbf{z} + \delta_{\mathbf{x}})] - T_{\mathbf{x}}^T I_{\mathbf{x}})^2 - h(\delta_{\mathbf{x}}), \tag{1}$$

where $h(\cdot)$ is a regularization term for the perturbations. The square loss promotes the robustness towards perturbation and the regularization ensures the perturbation is imperceptible. To avoid being stuck in the non-trivial stationary point, i.e. the original pre-trained parameters, we propose to add Gaussian Smoothing to the original function. This can also improve the smoothness of the loss term.

We observe the first-order Taylor approximation of Eq. 1 can be formulated as follows:

$$\min_\theta \mathbb{E}_{\mathbf{x} \sim \mathcal{D}_{\text{train}}} m_{\mathbf{x}}(\mathbf{0}) + \max_{\delta_{\mathbf{x}}} \delta_{\mathbf{x}}^T \omega_{\mathbf{x}} \nabla_{\mathbf{x}} \mathbb{E}_{\mathbf{z} \sim \mathcal{N}(\mathbf{0}, \sigma^2 \mathbb{I})}[T_{\mathbf{x}}^T f_\theta(\mathbf{x} + \mathbf{z})] - h(\delta_{\mathbf{x}}), \tag{2}$$

where $m_{\mathbf{x}}(\delta_{\mathbf{x}}) = \frac{1}{2}(T_{\mathbf{x}}^T \mathbb{E}_{\mathbf{z} \sim \mathcal{N}(\mathbf{0}, \sigma^2 \mathbb{I})}[f_\theta(\mathbf{x} + \mathbf{z} + \delta_{\mathbf{x}})] - T_{\mathbf{x}}^T I_{\mathbf{x}})^2$ and $\omega_{\mathbf{x}} = |T_{\mathbf{x}}^T \mathbb{E}_{\mathbf{z} \sim \mathcal{N}(\mathbf{0}, \sigma^2 \mathbb{I})}[f_\theta(\mathbf{x} + \mathbf{z})] - T_{\mathbf{x}}^T I_{\mathbf{x}}|$. The following theorem shows the optimized $\delta_{\mathbf{x}}^*$ in Eq. 1 and Eq. 2 are close to each other and their relative error is bounded for any $\mu$-strongly-convex $h(\cdot)$.

**Definition 1.** *We call $\varphi : \mathbb{R}^d \to \mathbb{R}$ $\mu$-strongly-convex if for every $\mathbf{x}$, $\mathbf{z}$ and $t \in [0, 1] : \varphi(t\mathbf{x} + (1 - t)\mathbf{z}) \leq t\varphi(\mathbf{x}) + (1 - t)\varphi(\mathbf{z}) - \frac{1}{2}\mu t(1 - t)\|\mathbf{z} - \mathbf{x}\|^2$. (**Remarks:** $\varphi$ is not necessarily differentiable.)*

**Theorem 1.** *Assume $h(\cdot)$ is $\mu$-strongly-convex. For every $\theta$ and $\mathbf{x}$, define $\delta_{1\theta}^*(\mathbf{x})$ to be the optimal solution in Eq. 1 and $\delta_{2\theta}^*(\mathbf{x})$ to be the optimal solution in Eq. 2. Then, their relative difference satisfies the following:*

$$\|\delta_{1\theta}^*(\mathbf{x}) - \delta_{2\theta}^*(\mathbf{x})\| / \|\delta_{1\theta}^*(\mathbf{x})\| \leq 5/(2\mu\sigma^2). \tag{3}$$

We defer the proof to the Appendix A.1. The bound shows that with reasonable $\mu\sigma^2$, $\delta_{1\theta}^*(\mathbf{x})$ and $\delta_{2\theta}^*(\mathbf{x})$ would be close, and therefore Eq. 2 is a good approximation of Eq. 1. Moreover, with $h^\star(\cdot)$ to be the Fenchel conjugate of $h(\cdot)$, we can rewrite Eq. 2 as: $\min_\theta \mathbb{E}_{\mathbf{x} \sim \mathcal{D}_{\text{train}}} m_{\mathbf{x}}(\mathbf{0}) + h^\star(\omega_{\mathbf{x}} \nabla_{\mathbf{x}} \mathbb{E}_{\mathbf{z} \sim \mathcal{N}(\mathbf{0}, \sigma^2 \mathbb{I})}[T_{\mathbf{x}}^T f_\theta(\mathbf{x} + \mathbf{z})])$. We notice the term $\nabla_{\mathbf{x}} \mathbb{E}_{\mathbf{z} \sim \mathcal{N}(\mathbf{0}, \sigma^2 \mathbb{I})}[T_{\mathbf{x}}^T f_\theta(\mathbf{x} + \mathbf{z})]$ is actually the SmoothGrad (Smilkov et al., 2017) of similarity score w.r.t input. Therefore, we conclude the proposed AFT has an implicit regularization on the input gradients, enforced by the function $h^\star(\cdot)$.

Following this general guideline, we can identify an appropriate $h^\star(\cdot)$ to enforce sparse concept encoding of the image encoder while simultaneously regulating the approximation error. We propose to regularize the SmoothGrad with the smoothed version of $L_1$-norm regularization:

$$h^\star(\mathbf{u}) = \epsilon \sum_i H_\eta(\mathbf{u}_{(i)}), \text{ where } H_\eta(\mathbf{u}_{(i)}) = \begin{cases} \frac{1}{2\eta}\mathbf{u}_{(i)}^2, & \text{if } |\mathbf{u}_{(i)}| \leq \eta \\ |\mathbf{u}_{(i)}| - \frac{\eta}{2}, & \text{otherwise}, \end{cases} \tag{4}$$

and the corresponding $h(\cdot)$ is strongly-convex and can be written as:

$$h(\mathbf{v}) = \begin{cases} \frac{\eta}{2\epsilon}\|\mathbf{v}\|^2, & \text{if } \|\mathbf{v}\|_\infty \leq \epsilon \\ +\infty, & \text{otherwise}, \end{cases} \tag{5}$$

where $\epsilon, \eta > 0$. $\epsilon$ is a hyper-parameter controlling the strength of the regularization and $\eta$ determines the smoothness of $h^\star(\cdot)$ and therefore the convexity of $h(\cdot)$. This sparsity-inducing regularization forces the visual encoder to focus on only a few key features from the input samples. For one thing, it can mitigate the noisy issue of the saliency maps, making them of higher visual quality. For the other, it can boost the conciseness during the concept encoding phase of the network and potentially make the generated concept architecture-agnostic (Ren et al., 2023). Overall, it makes the reasoning process of the visual encoder more interpretable.

We would highlight that although we enforce sparsity in this particular application, our framework is highly flexible. We can switch $h^\star(\cdot)$ to other smooth regularization (e.g. smoothed version of group-norm or elastic net regularization) to enforce different properties of the encoder (see Appendix A.4).

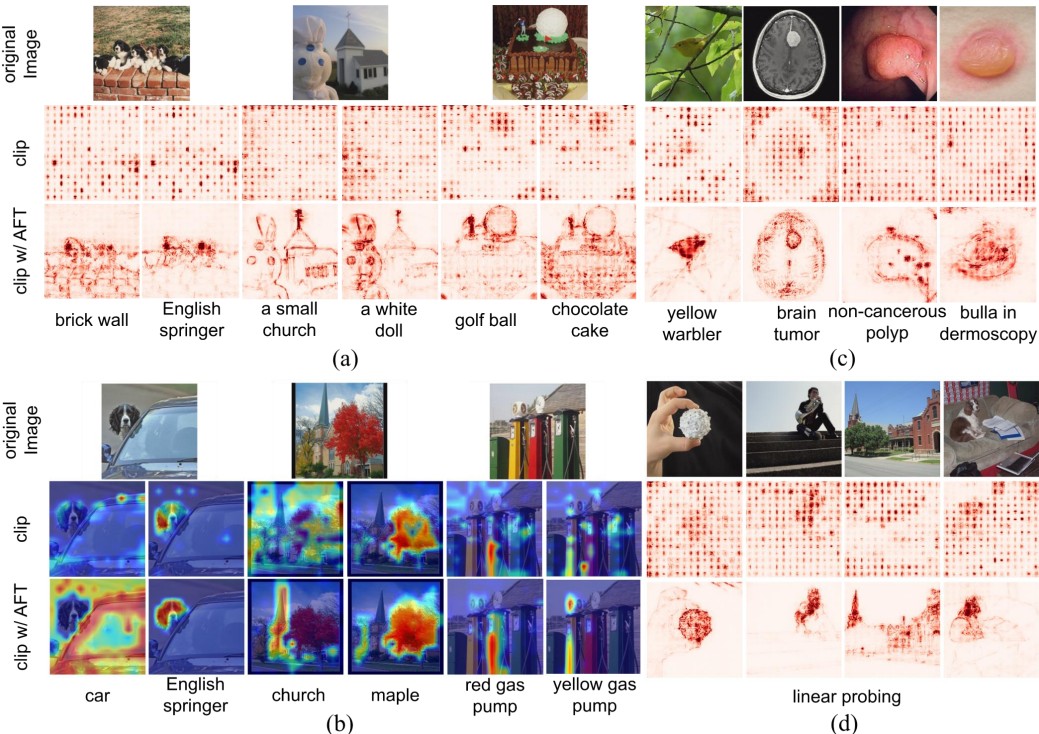

Figure 2: (a,b) Comparison of Simple Gradients/Grad-Cam between CLIP w/wo AFT. AFT greatly improves the visual quality. (c) Evaluation of Simple Gradients on out-of-distribution dataset. (d) Evaluation of Simple Gradients with linear probing. The improvements of visual interpretability stem from AFT can transfer across datasets and to different tasks.

## 3.2 UNSUPERVISED TRAINING BY UPPER BOUND MINIMIZATION

One limitation of the objective function in Eq. 1 is that it necessitates the calculation of $T_{\mathbf{x}}$. This makes the fine-tuning process computationally expensive, as we still need the text embedding encoded by the text encoder, despite the text encoder being frozen. Furthermore, this constraint necessitates that the fine-tuning process be conducted on datasets containing language-image pairs, thereby limiting the potential to fine-tune the encoder on image-only datasets, which are more readily available in many applications, such as the medical imaging domain. Additionally, the performance is sensitive to the quality of the text descriptions. Fine-tuning on small scale image-text pairs can degenerate the zero-shot ability (Mao et al., 2022). However, we have observed that the inner maximization objective of Eq. 1 has a uniform upper bound independent of the text embedding:

$$\max_{\delta_{\mathbf{x}}} m_{\mathbf{x}}(\delta_{\mathbf{x}}) - h(\delta_{\mathbf{x}}) \leq \max_{\delta_{\mathbf{x}}} \frac{1}{2}\|\mathbb{E}_{\mathbf{z}\sim\mathcal{N}(\mathbf{0},\sigma^2\mathbb{I})}[f_\theta(\mathbf{x} + \mathbf{z} + \delta_{\mathbf{x}})] - I_{\mathbf{x}}\|^2 - h(\delta_{\mathbf{x}}). \tag{6}$$

Therefore, we can minimize this upper bound instead to bypass the demand for text embeddings:

$$\min_{\theta} \mathbb{E}_{\mathbf{x}\sim\mathcal{D}_{\text{train}}} \max_{\delta_{\mathbf{x}}} \frac{1}{2}\|\mathbb{E}_{\mathbf{z}\sim\mathcal{N}(\mathbf{0},\sigma^2\mathbb{I})}[f_\theta(\mathbf{x} + \mathbf{z} + \delta_{\mathbf{x}})] - I_{\mathbf{x}}\|^2 - h(\delta_{\mathbf{x}}). \tag{7}$$

The optimization objective is now the $L_2$-distance between the perturbed image embedding and the original image embedding. In the Appendix A.4, we empirically study the effects of this approximation. Moreover, we have the following observation showing after this unsupervised fine-tuning, the smoothed image embedding exhibits alignment with the original text embedding.

**Observation 1.** *For every image $\mathbf{x}$, its original image embedding $I_{\mathbf{x}}$, embedding of text prompt $T_{\mathbf{x}}$, and fine-tuned network parameters $\theta$, if $\|\mathbb{E}_{\mathbf{z}\sim\mathcal{N}(\mathbf{0},\sigma^2\mathbb{I})}[f_\theta(\mathbf{x} + \mathbf{z})] - I_{\mathbf{x}}\| \leq \lambda$, then $|T_{\mathbf{x}}^T\mathbb{E}_{\mathbf{z}\sim\mathcal{N}(\mathbf{0},\sigma^2\mathbb{I})}[f_\theta(\mathbf{x} + \mathbf{z})] - T_{\mathbf{x}}^T I_{\mathbf{x}}| \leq \lambda.$*

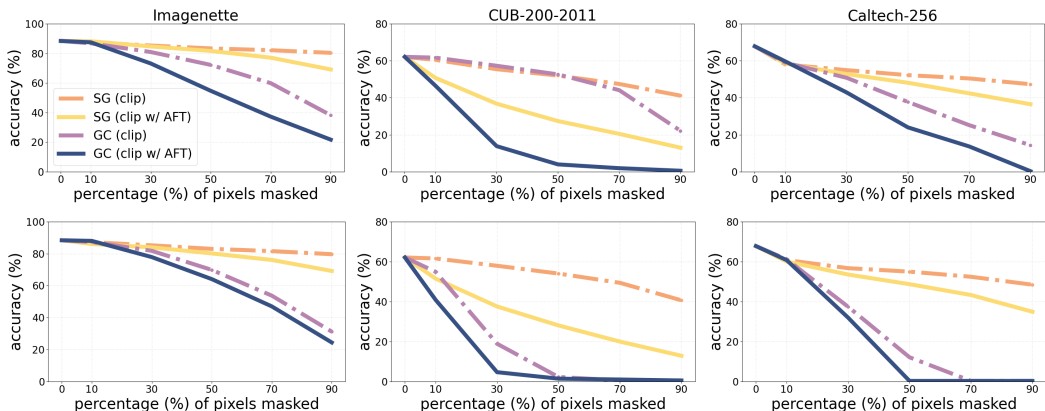

Figure 3: Analysis of Remove and Retrain (ROAR) on zero-shot classification (**top**) and linear probing (**bottom**). A more rapid drop in accuracy shows the highlighted features are more important. AFT enables CLIP's vision encoder to better capture task-related features.

Therefore, the smoothed image embedding is adaptable to the CLIP's text encoder. Empirically, We discover $f_\theta(\mathbf{x})$ is close to $\mathbb{E}_{\mathbf{z}\sim\mathcal{N}(\mathbf{0},\sigma^2\mathbb{I})}[f_\theta(\mathbf{x}+\mathbf{z})]$, since $\sigma$ will not be too large. Therefore $f_\theta(\mathbf{x})$ can also be compatible with the text encoder, and potentially downstream large vision-language models tuned using the CLIP's embeddings. Note the framework can be generalized to any foundation model that entails an intermediate embedding layer linking modalities. We will show numerical results on applying this framework to multi-modal models other than CLIP in the Appendix A.4

## 4 EXPERIMENTS

We fine-tuned the CLIP using the ImageNet (Deng et al., 2009) training set for 2 epochs. We experimented with both ViT (Dosovitskiy et al., 2020) and ResNet (He et al., 2016) architecture. Without special illustration, the results are based on ViT-B/16 with $\sigma = \eta = 1/255$ and $\epsilon = 4/255$. More comprehensive results can be found in the Appendix A.4. For AFT, we applied PGD (Madry et al., 2017) for 10 steps. All experiments were conducted on NVIDIA GeForce RTX 4090 GPUs.

### 4.1 IMPROVEMENTS OF SALIENCY MAPS THROUGH AFT

We first examine how AFT can improve the quality of the saliency maps. Our experiments primarily focus on two most fundamental saliency maps: *Simple Gradients* (SG) and *Grad-Cam* (GC). Both methods rely on unaltered gradient information for local explanation, eschewing complex design and post-processing aimed at quality enhancement. We believe this can better reflect the inherent interpretability of the network. Superior saliency maps should be indicative of an improved interpretability of the network, rather than the result of intricate manipulations of the saliency maps.

**Visual Quality.** We visualize several saliency maps for images selected from the validation set of ImageNet in Fig. 2 (a). The SG of the original CLIP looks largely stochastic without highlighting any meaningful patterns. It also exhibits minimal variation when different prompts are used. Conversely, after AFT, the SG demonstrates substantial improvements, accurately capturing the objects within the image related to the prompt. It also displays increased sensitivity to the prompt variations. This trend is similarly observed for GC (Fig. 2 (b)). Although the quality of GC for the original CLIP is superior to SG, it still suffers from background noise interference. AFT enables the GC to align better with the objects in the foreground. The results show that the original CLIP would rely on patterns that are obscure to human perception for decision-making. After AFT, the model leverages more human-interpretable concepts for prediction, thereby enhancing its interpretability.

**Transferability.** We then investigate the transferability of the improvements in interpretability to diverse datasets and downstream tasks. We visualize the SG for several images sourced from fine-grained classification and medical image datasets (Fig. 2 (c)), which are not included in the AFT phase and exhibit domain gaps with the training data. Moreover, their corresponding labels are absent from the ImageNet dataset. Nevertheless, the SG continues to exhibit meaningful patterns on

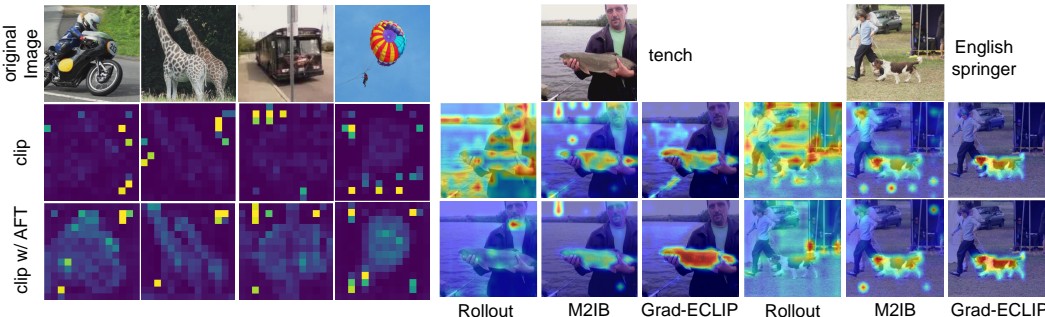

Figure 4: **Left**: Visualization of attention maps of ViT. AFT helps produce more fine-grained features. **Right**: Visualization of saliency maps produced by Rollout, M2IB, and Grad-ECLIP. AFT can consistently improve the quality of saliency maps.

Table 1: Evaluation of localization ability using the Point Game (PG and PG-energy) and Segmentation test (Pix. Acc., AP and MaskIoU) on the ImageNet-Segmentation validation dataset.

| Tasks | | Zero-shot Classification | | | | | Linear Probing | | | | |
|---|---|---|---|---|---|---|---|---|---|---|---|
| Saliency maps | CLIP | PG↑ | PG-energy↑ | Pixel Acc.↑ | AP↑ | mask-IoU↑ | PG↑ | PG-energy↑ | Pixel Acc.↑ | AP↑ | mask-IoU↑ |
| SG | w/o AFT | 25.77 | 31.32 | 65.51 | 34.45 | 2.16 | 27.75 | 32.62 | 65.45 | 35.22 | 2.51 |
| | w/ AFT | **71.15** | **53.98** | **66.67** | **59.85** | **4.31** | **65.79** | **51.20** | **66.53** | **56.52** | **4.06** |
| GC | w/o AFT | 51.00 | 43.71 | 61.79 | 49.18 | 16.19 | 33.29 | 36.28 | **60.55** | 38.85 | 9.89 |
| | w/ AFT | **73.98** | **63.76** | **67.44** | **62.34** | **19.39** | **33.59** | **37.02** | 59.20 | **40.33** | **10.62** |
| Grad-ECLIP | w/o AFT | 86.61 | 60.54 | 71.19 | 76.82 | 25.70 | NA | NA | NA | NA | NA |
| | w/ AFT | **87.40** | **62.77** | **72.23** | **78.34** | **27.97** | NA | NA | NA | NA | NA |

these out-of-distribution images. We further conduct linear probing as a common downstream task of CLIP and visualize the SG for the resulting models. In Fig. 2 (d), the SG maintains high quality. This shows that the improved visual interpretability can be transferred to downstream applications.

**Feature Importance.** To quantitatively analyze whether the saliency maps highlight the task-related regions of the input image, we conduct Remove and Retrain (ROAR) analysis (Hooker et al., 2019) on the saliency maps. ROAR reflects the drop in the predictive power of the dataset when increasing the proportion of pixels removed according to their importance score in the saliency maps. A steeper decline in performance indicates that the model effectively captures task-related features from the inputs. We retrain networks with the top $k\%$ of the pixels removed from each image and record the accuracy on the test set. Each measurement is based on three trials with random initialization. We perform the analysis on both in-distribution datasets (Imagenette, a ten-class subset of ImageNet) and out-of-distribution datasets (CUB-200 (Wah et al., 2011) and Caltech-256 (Griffin et al., 2007)). We apply ROAR on saliency maps generated from both zero-shot prediction and liner probing. The results in Fig. 3 show an obvious pattern that the drop is more rapid when the CLIP is adversarially fine-tuned. It holds for both SG and GC, showing AFT enables CLIP to focus on important features.

**Improvements on Raw Attention Maps and Advanced Saliency Maps.** We visualize the last attention maps of the ViT in Fig. 4, which illustrate the attention score between the $[cls]$ token and other image tokens. It is evident that after AFT, the attention maps show more fine-grained features, indicating that the model can better capture the details from the input image. We further showcase several saliency maps generated by more recent feature attribution methods, including techniques designed for transformer-based models such as Rollout (Abnar & Zuidema, 2020), and state-of-the-art CLIP interpretation techniques like M2IB (Wang et al., 2023) and Grad-ECLIP (Zhao et al., 2024). With modern explanation techniques, we observe improved saliency maps even when using the original CLIP visual encoder. However, AFT consistently enhances the visual quality of the saliency maps, making the saliency maps sparser and better aligned with the object described in the text prompts. Therefore, the improvements of AFT are explanation-agnostic.

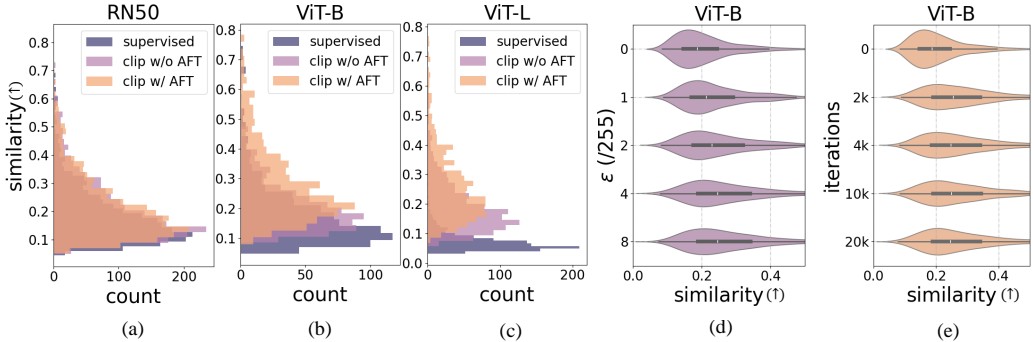

(a)    (b)    (c)    (d)    (e)

Figure 5: (a-c) Comparison of neuron-concept alignment w/wo AFT via CLIP-dissect. (d,e) Effects of perturbation strength/tuning iterations on neuron-concept alignment.

**Pointing Game.** Finally, we assess the localization capabilities of the saliency maps using the ImageNet-Segmentation (Gao et al., 2022) validation set, which includes segmentation annotations for 12,419 images across 919 categories from ImageNet. We use the Point Game (PG) as a standard metric to measure the accuracy of visual explanation localization. The PG metric assigns a hit score if the point with the highest value on the text-specific heat map falls within the object region, defined by the class segmentation mask. The PG accuracy is then calculated by averaging all sample scores. Additionally, we employ the energy-PG (Wang et al., 2020), which measures the ratio of heat map energy within the ground truth mask to the entire map, providing a better reflection of the heat map's spread. Following Chefer et al. (2021a;b), we treat the heat maps as soft-segmentation results and calculate pixel accuracy (Pixel Acc.), average precision (AP), and averaged mask intersection over union (maskIoU) for evaluation. The results are shown in Table 6, which cover evaluation for both SG, GC and one advanced saliency map technique, i.e., Grad-ECLIP. The evaluation is based on saliency maps generated by both zero-shot classification and linear probing. For most of the metrics, AFT brings improvement over the original model. For example, the PG of zero-shot classification improves by 45.38% and 22.98% for SG and GC respectively. The results show that AFT makes the saliency maps better localize at the object with the correct category as the text.

### 4.2 INTERPRETABILITY QUANTIFICATION VIA NETWORK DISSECTION

To evaluate the interpretability of the visual encoder, we further conduct network dissection and evaluate the quantity and quality of the concept detector within the visual encoder. The concept detector refers to the neuron that exclusively responds to specific concepts understandable to human beings. We apply CLIP-dissect (Oikarinen & Weng, 2022) to discover the concept detectors. Specifically, we use the the broadly and densely labeled (Broden) dataset (Bau et al., 2017) as a probing dataset, which covers comprehensive human-understandable concepts including objects, scenes, parts, textures, materials, and colors. Then we have a concept set comprised of 20,000 most common English words. For each concept $c$, we calculate its similarity score with all the images within the probing dataset using a pre-trained CLIP model, which results in a similarity array $\mathbb{S}_c$. For a specific neuron $k$, we calculate its activation value with each image in the probing dataset as input, which also generates an array $\mathbb{A}_k$. The alignment between neuron $k$ and concept $c$ is measured by comparing the similarity between $\mathbb{A}_k$ and $\mathbb{S}_c$. The neuron $k$ is called a concept detector of concept $c_k$ if $c_k$ has the largest alignment with neuron $k$ among all the concepts within the concept set. Moreover, the larger the similarity is, the better alignment is achieved between the neuron and the concept.

**Improvements over Original CLIP.** We first show that AFT can improve the interpretability of the CLIP visual encoder. To achieve this, we evaluate the alignment score of each neuron in the $[cls]$ token of the last layer with its corresponding concept and visualize the scores via histogram. For comparison, we draw the plots for CLIP w/o AFT, as well as for a network trained using supervised learning on the ImageNet dataset. Additionally, we conduct experiments on three different CLIP architectures: ResNet-50 (He et al., 2016), ViT-B, and ViT-L (Dosovitskiy et al., 2020). The results are shown in Fig. 5. The data reveals a clear trend indicating that CLIP exhibits superior alignment with concepts compared to supervised learning, corroborating previous findings by (Goh et al.,

2021), which demonstrated that neurons in CLIP align more effectively with specific words due to the model's objective of aligning image representations with text representations. Moreover, our proposed AFT can further improve the alignments. AFT mitigates the non-smoothness introduced by the false negative samples during contrastive learning, thereby promoting the alignment between image representations and their corresponding concepts. This trend can be more pronounced as the complexity of the network architecture increases.

We then conduct Network Dissection (Bau et al., 2017) on the penultimate layer of ViT-B and visualize the activation map of a certain 'cat' detector neuron towards several top activated probing images in Fig. 6. The results also illustrate that the activation map of the model after AFT could be more object-centric.

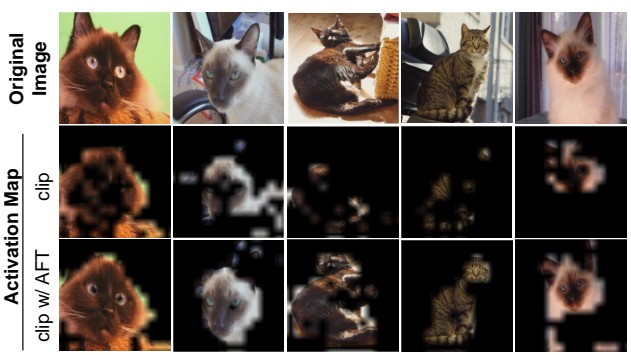

Figure 6: Comparison between activation maps of the same cat detector. The detector corresponds to the neuron with the highest IoU score with samples belonging to the cat concept.

**Effects of Hyper-Parameters.** We further study the relationship between training hyper-parameters and the model interpretability. Specifically, we study the influence of adversarial perturbation strength and fine-tuning iterations. According to the results in Fig. 5 (d), the alignment with the concept improves with stronger adversarial perturbations. However, it stops further improving when $\epsilon$ becomes greater than 4/255. In Fig. 5 (e), we find that the concept alignment can be improved with only a small amount of tuning iterations, showing the efficiency of the proposed AFT scheme.

## 4.3 EXTENDED EXPERIMENTS

**Interpretability Improvements for Large Vision-Language Models.** As the pre-trained CLIP is an integral component for many large vision-language models (e.g., LLaVa (Liu et al., 2024)), we postulate the improvements of the CLIP visual interpretability can be propagated to downstream vision-language models. To substantiate this, we visualize the attention maps of LLaVa, corresponding to various tokens from the generated sentences. Specifically, we integrate the attention weights of the LLM with those of the ViT, which produces a composite attention map over the input image. We compare the attention maps of the LLaVa equipped with the original CLIP visual encoder and the one after AFT. We show several results from the COCO dataset (Lin et al., 2014) in Fig. 7. The attention maps with AFT show much cleaner saliency maps, prominently and sparsely highlighting the objects corresponding to the specific tokens. This indicates that interpretability improvements can indeed be transferred to vision-language models without compromising the quality of the generated sentences. The interpretation can also help explain some of the hallucinations. For example, in the third image of Fig. 7, the interpretation map implies the model mistakes the black bag as a cat. In the Appendix A.4, we show quantitatively how AFT benefits the downstream VLMs' interpretability.

**Trade-Off with Zero-Shot Accuracy.** Finally, we analyze the trade-off between interpretability and zero-shot accuracy. Following the same benchmark with Mao et al. (2022), we assess classification accuracy on the ImageNet test set and 15 zero-shot recognition tasks, ensuring no overlapping with the fine-tuning dataset. We report both clean accuracy and robust accuracy under the first two attacks of AutoAttack (Croce & Hein, 2020), i.e., APGD with cross-entropy and DLR loss. We use the ViT-L/14 version of the CLIP to be compatible with previous work (Schlarmann et al., 2024). The results are shown in Table 2 (and the complete results are in Appendix A.4). The results show a certain drop in clean accuracy. However, a minimal $\epsilon$ like 1/255 can result in slight accuracy decrement while still substantially enhancing interpretability (see Appendix A.4). Beyond interpretability, AFT can also markedly improve the robustness of the CLIP model, which is another crucial factor to consider in high-stake applications. Furthermore, our experiments involving AFT on CLIP with the ImageNet training set reveal a smaller accuracy drop for ImageNet compared to other datasets. This suggests that the trade-off can be further mitigated by AFT the CLIP with more diverse data.

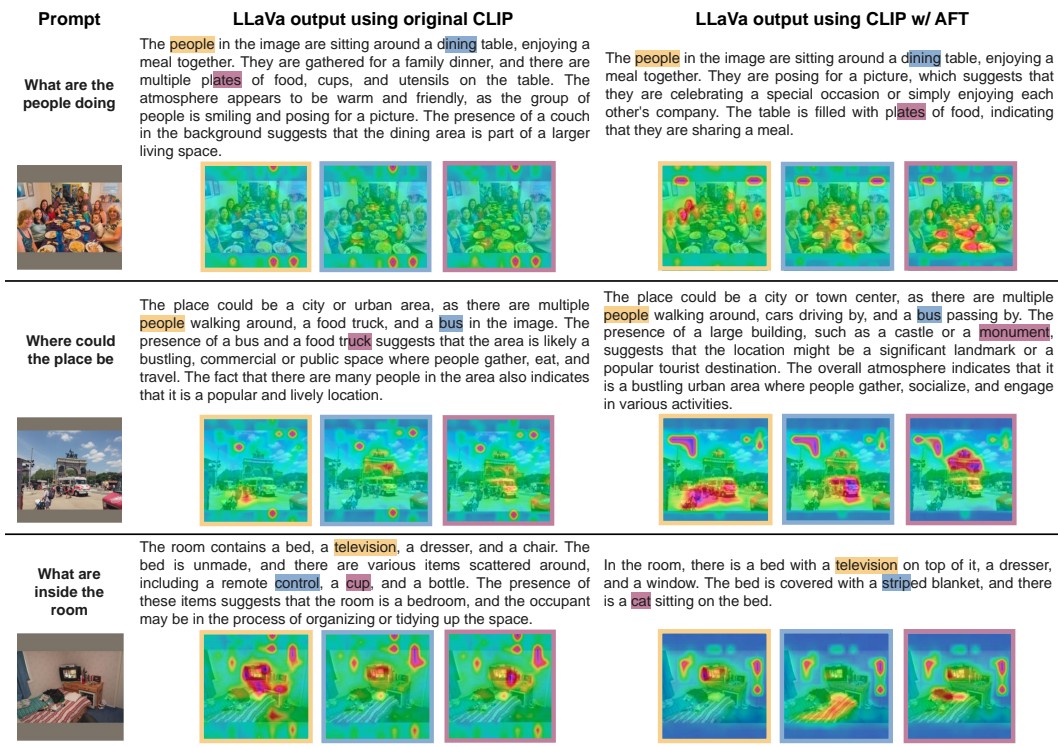

Figure 7: Comparison of interpretation maps for LLaVa with the visual encoder replaced by CLIP visual encoder w/wo AFT. The attention maps correspond to the tokens highlighted with the same color as the border of the attention maps.

Table 2: Accuracy evaluation on image classification datasets of CLIP model. The superscripts on the upper right side show the value of $\epsilon$ (/255) during the adversarial fine-tuning.

| $L_\infty$ | Vision Encoder | ImageNet | CIFAR10 | STL-10 | CIFAR100 | Cars | CalTech | OxfordPets | Flowers | DTD | EuroSAT | FGVC | PCAM | ImageNet-R | ImageNet-S | Average Zero-shot |
|---|---|---|---|---|---|---|---|---|---|---|---|---|---|---|---|---|
| clean | CLIP | 74.90 | 95.20 | 99.31 | 71.08 | 77.94 | 83.30 | 93.21 | 79.17 | 55.21 | 62.65 | 31.77 | 52.00 | 87.86 | 59.59 | 72.95 |
| | AFT[1] | 75.92 | 94.14 | 99.11 | 75.34 | 74.23 | 84.47 | 92.91 | 75.74 | 54.47 | 31.61 | 28.71 | 52.70 | 87.56 | 60.42 | 70.11 |
| | AFT[2] | 74.60 | 89.71 | 98.50 | 69.66 | 70.15 | 85.03 | 91.09 | 70.34 | 50.00 | 24.74 | 27.57 | 50.02 | 85.45 | 59.58 | 67.06 |
| | AFT[4] | 70.88 | 78.47 | 96.28 | 57.31 | 63.36 | 84.73 | 86.97 | 57.64 | 42.93 | 18.56 | 22.35 | 50.02 | 80.39 | 56.91 | 60.99 |
| 2/255 | CLIP | 0.00 | 0.00 | 0.00 | 0.00 | 0.00 | 0.00 | 0.00 | 0.00 | 0.00 | 0.00 | 0.00 | 0.00 | 0.00 | 0.10 | 0.00 |
| | AFT[2] | 47.36 | 61.70 | 90.80 | 37.40 | 25.50 | 73.80 | 68.60 | 31.70 | 26.50 | 8.60 | 5.90 | 46.90 | 57.40 | 38.70 | 44.12 |
| | AFT[4] | 53.68 | 57.90 | 90.20 | 38.00 | 30.70 | 77.60 | 72.50 | 30.20 | 28.80 | 12.60 | 8.00 | 50.20 | 62.10 | 43.10 | 46.30 |
| 4/255 | CLIP | 0.00 | 0.00 | 0.00 | 0.00 | 0.00 | 0.00 | 0.00 | 0.00 | 0.00 | 0.00 | 0.00 | 0.00 | 0.00 | 0.00 | 0.00 |
| | AFT[2] | 18.34 | 26.90 | 63.40 | 14.30 | 5.40 | 47.30 | 30.50 | 6.90 | 12.50 | 1.50 | 0.50 | 19.70 | 27.00 | 23.10 | 21.46 |
| | AFT[4] | 35.28 | 36.40 | 75.70 | 21.20 | 12.80 | 65.90 | 51.50 | 13.00 | 17.60 | 11.30 | 2.60 | 50.20 | 41.00 | 31.60 | 33.14 |

## 5 LIMITATIONS AND CONCLUSION

In this work, we propose an unsupervised adversarial fine-tuning scheme for improving the visual interpretability of the CLIP. We provide theoretical analysis to explain the underlying mechanisms driving these improvements. Through comprehensive quantitative and qualitative evaluations, we demonstrate the proposed method enables the CLIP visual encoder to focus more effectively on salient input features, and the neurons exhibit improved alignment with human-understandable concepts. Moreover, the effects are both generalizable and transferable.

Our work is subject to certain limitations. We focus on fine-tuning with a single datasets, without investigating the impact of data quantity and data diversity on performance. Additionally, we have not explored the effects of different minimax optimization algorithms. Our future work entails more comprehensive studies regarding these aspects.

ACKNOWLEDGMENTS

The work of Farzan Farnia is partially supported by a grant from the Research Grants Council of the Hong Kong Special Administrative Region, China, Project 14209920, and is partially supported by CUHK Direct Research Grants with CUHK Project No. 4055164 and 4937054. This work was also supported in part by a grant from the Research Grants Council of the Hong Kong Special Administrative Region, China (Project No. T45-401/22-N). Finally, the authors would like to thank the anonymous reviewers for their insightful suggestions and feedback.

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

# A  APPENDIX

In this Appendix, we provide proof for the theorems, implementation details, additional experimental results, and a discussion on the potential application of a more interpretable CLIP.

## A.1  PROOFS

**Proof of Theorem 1.**    To prove this theorem, we apply the multivariate version of Stein's lemma by Landsman et al. (2013), which shows that for a bounded function $g : \mathbb{R}^d \to \mathbb{R}, |g\| \le M$ and Gaussian noise $\mathbf{z} \sim \mathcal{N}(\mathbf{0}, \sigma^2 \mathbb{I})$, $\mathbb{E}_{\mathbf{z} \sim \mathcal{N}(\mathbf{0}, \sigma^2 \mathbb{I})}[g(\mathbf{x} + \mathbf{z})]$ is $\frac{M}{\sigma^2}$-smooth. Therefore, $\mathbb{E}_{\mathbf{z} \sim \mathcal{N}(\mathbf{0}, \sigma^2 \mathbb{I})}[T_{\mathbf{x}}^T f_\theta(\mathbf{x} + \mathbf{z})]$ is $\frac{1}{\sigma^2}$-smooth, and $\frac{1}{2}(T_{\mathbf{x}}^T \mathbb{E}_{\mathbf{z} \sim \mathcal{N}(\mathbf{0}, \sigma^2 \mathbb{I})}[f_\theta(\mathbf{x} + \mathbf{z})] - T_{\mathbf{x}}^T I_{\mathbf{x}})^2$ is $\frac{5}{2\sigma^2}$-smooth.

According to the properties of smooth function, we have:

$$m_{\mathbf{x}}(0) + \max_\delta \delta^T \omega_{\mathbf{x}} \nabla_{\mathbf{x}} \mathbb{E}_{\mathbf{z} \sim \mathcal{N}(\mathbf{0}, \sigma^2 \mathbb{I})}[T_{\mathbf{x}}^T f_\theta(\mathbf{x} + \mathbf{z})] - (h(\delta) + \frac{5}{4\sigma^2}\|\delta\|^2) \le \max_\delta m_{\mathbf{x}}(\delta) - h(\delta)$$

$$\le m_{\mathbf{x}}(0) + \max_\delta \delta^T \omega_{\mathbf{x}} \nabla_{\mathbf{x}} \mathbb{E}_{\mathbf{z} \sim \mathcal{N}(\mathbf{0}, \sigma^2 \mathbb{I})}[T_{\mathbf{x}}^T f_\theta(\mathbf{x} + \mathbf{z})] - (h(\delta) - \frac{5}{4\sigma^2}\|\delta\|^2).$$

$$(8)$$

We reparameterize the obj. 1 as:

$$\max_\delta \delta^T \omega_{\mathbf{x}} \nabla_{\mathbf{x}} \mathbb{E}_{\mathbf{z} \sim \mathcal{N}(\mathbf{0}, \sigma^2 \mathbb{I})}[T_{\mathbf{x}}^T f_\theta(\mathbf{x} + \mathbf{z})] - (h(\delta) + \xi\|\delta\|^2), \tag{9}$$

where $-\frac{5}{4\sigma^2} \le \xi \le \frac{5}{4\sigma^2}$. By definition, we have:

$$\delta_{1\theta}^*(\mathbf{x}) := \arg\max_\delta \delta^T \omega_{\mathbf{x}} \nabla_{\mathbf{x}} \mathbb{E}_{\mathbf{z} \sim \mathcal{N}(\mathbf{0}, \sigma^2 \mathbb{I})}[T_{\mathbf{x}}^T f_\theta(\mathbf{x} + \mathbf{z})] - (h(\delta) + \xi\|\delta\|^2), \tag{10}$$

$$\delta_{2\theta}^*(\mathbf{x}) := \arg\max_\delta \delta^T \omega_{\mathbf{x}} \nabla_{\mathbf{x}} \mathbb{E}_{\mathbf{z} \sim \mathcal{N}(\mathbf{0}, \sigma^2 \mathbb{I})}[T_{\mathbf{x}}^T f_\theta(\mathbf{x} + \mathbf{z})] - h(\delta). \tag{11}$$

As $\delta_{1\theta}^*(\mathbf{x})$ and $\delta_{2\theta}^*(\mathbf{x})$ are stationary points of the corresponding objectives, we have:

$$\mathbb{E}_{\mathbf{z} \sim \mathcal{N}(\mathbf{0}, \sigma^2 \mathbb{I})}[T_{\mathbf{x}}^T f_\theta(\mathbf{x} + \mathbf{z})] - \nabla_\delta h(\delta_{1\theta}^*(\mathbf{x})) - 2\xi(\mathbf{x}) = 0, \tag{12}$$

$$\mathbb{E}_{\mathbf{z} \sim \mathcal{N}(\mathbf{0}, \sigma^2 \mathbb{I})}[T_{\mathbf{x}}^T f_\theta(\mathbf{x} + \mathbf{z})] - \nabla_\delta h(\delta_{2\theta}^*(\mathbf{x})) = 0. \tag{13}$$

Therefore, we have:

$$\nabla_\delta h(\delta_{1\theta}^*(\mathbf{x})) - \nabla_\delta h(\delta_{2\theta}^*(\mathbf{x})) = 2\xi \delta_{1\theta}^*(\mathbf{x}). \tag{14}$$

Moreover, since $h(\cdot)$ is $\mu$-strongly-convex, we further have:

$$(\nabla_\delta h(\delta_{1\theta}^*(\mathbf{x})) - \nabla_\delta h(\delta_{2\theta}^*(\mathbf{x})))^T (\delta_{1\theta}^*(\mathbf{x}) - \delta_{2\theta}^*(\mathbf{x})) \ge \mu\|\delta_{1\theta}^*(\mathbf{x}) - \delta_{2\theta}^*(\mathbf{x})\|^2. \tag{15}$$

Meanwhile, according to the Cauchy–Schwarz inequality, we have:

$$(\nabla_\delta h(\delta_{1\theta}^*(\mathbf{x})) - \nabla_\delta h(\delta_{2\theta}^*(\mathbf{x})))^T (\delta_{1\theta}^*(\mathbf{x}) - \delta_{2\theta}^*(\mathbf{x}))$$
$$\le \|\nabla_\delta h(\delta_{1\theta}^*(\mathbf{x})) - \nabla_\delta h(\delta_{2\theta}^*(\mathbf{x}))\|\|\delta_{1\theta}^*(\mathbf{x}) - \delta_{2\theta}^*(\mathbf{x})\|. \tag{16}$$

Combining 16 and 15: We have:

$$\mu\|\delta_{1\theta}^*(\mathbf{x}) - \delta_{2\theta}^*(\mathbf{x})\| \le \|\nabla_\delta h(\delta_{1\theta}^*(\mathbf{x})) - \nabla_\delta h(\delta_{2\theta}^*(\mathbf{x}))\| = 2|\xi|\|\delta_{1\theta}^*(\mathbf{x})\| \tag{17}$$

So we finally have:

$$\|\delta_{1\theta}^*(\mathbf{x}) - \delta_{2\theta}^*(\mathbf{x})\|/\|\delta_{1\theta}^*(\mathbf{x})\| \le \frac{2|\xi|}{\mu} \le \frac{5}{2\mu\sigma^2}. \tag{18}$$

This completes the proof.

**Proof of Observation 1.**    By Cauchy–Schwarz inequality, we have:

$$|T_{\mathbf{x}}^T \mathbb{E}_{\mathbf{z} \sim \mathcal{N}(\mathbf{0}, \sigma^2 \mathbb{I})}[f_\theta(\mathbf{x} + \mathbf{z}) - T_{\mathbf{x}}^T I_{\mathbf{x}}| \le \|T_{\mathbf{x}}\|\|\mathbb{E}_{\mathbf{z} \sim \mathcal{N}(\mathbf{0}, \sigma^2 \mathbb{I})}[f_\theta(\mathbf{x} + \mathbf{z})] - I_{\mathbf{x}}\| \tag{19}$$

Since $T_x$ is a unit-length representation, we have:

$$|T_{\mathbf{x}}^T \mathbb{E}_{\mathbf{z} \sim \mathcal{N}(\mathbf{0}, \sigma^2 \mathbb{I})}[f_\theta(\mathbf{x} + \mathbf{z}) - T_{\mathbf{x}}^T I_{\mathbf{x}}| \le \|\mathbb{E}_{\mathbf{z} \sim \mathcal{N}(\mathbf{0}, \sigma^2 \mathbb{I})}[f_\theta(\mathbf{x} + \mathbf{z})] - I_{\mathbf{x}}\| \le \lambda. \tag{20}$$

This completes the proof.

## A.2 CLARIFICATION ON THE GAUSSIAN SMOOTHING

The theoretical derivation involves Gaussian Smoothed term. The Gaussian smoothing ensures that the sum of the Gaussian smoothed loss and the negative of strongly-convex regularization penalty (dual norm to Huber loss) will be a concave function with a unique maximizer and therefore guarantees the convergence of gradient updates in solving the inner maximization. Although the outer minimization task over CLIP weights still remains a challenging non-convex optimization, through using the Gaussian smoothed loss function and the dual function to Huber loss, we are able to solve the inner optimization problem and converge to a solution of the primal optimization problem with the Huber loss penalty. In the experiments, We draw the Gaussian noise vectors independently for each training sample at every PGD iteration.

## A.3 IMPLEMENTATION DETAILS

**Adversarial Training setup.** All the models in the paper are trained on ImageNet (at resolution 224×224) for two epochs using 10 steps of PGD, with step size set to 1/255.We use AdamW (Loshchilov et al., 2017) optimizer with momenta coefficients $\beta_1$ and $\beta_2$ to be 0.9 and 0.95 respectively. The training was done with a cosine decaying learning rate schedule with a linear warm-up and a peak learning rate to be 1e-5. The weight decay is set to 1e-4 and the batch size is 128 for RN50 and ViT-B and 64 for ViT-L respectively. We sample only 1 image per interaction from the Gaussian distribution as a simplified implementation of Gaussian smoothing.

**Zero-Shot Evaluations.** We use the CLIP Benchmark[1] and OpenCLIP[2] (Cherti et al., 2023) protocol to evaluate the zero-shot performance. The evaluation datasets include: CalTech101 (Fei-Fei et al., 2004), StanfordCars (Krause et al., 2013), CIFAR10, CIFAR100 (Krizhevsky et al., 2009), DTD (Cimpoi et al., 2014), EuroSAT (Helber et al., 2019) FGVC Aircrafts (Maji et al., 2013), Flowers (Nilsback & Zisserman, 2008), ImageNet-R (Hendrycks et al., 2021), ImageNet-Sketch (Wang et al., 2019), PCAM (Veeling et al., 2018), OxfordPets (Parkhi et al., 2012), and STL-10 (Coates et al., 2011).We also test performance on the validation set of ImageNet (Deng et al., 2009).

We evaluate robustness on 1000 samples each and report clean accuracy for all samples of the respective dataset. The attacks applied are the first two attacks of AutoAttack (Croce & Hein, 2020), i.e., APGD with cross-entropy loss and APGD with targeted DLR loss (100 iterations each). We consider $L_\infty$-bounded threat models with radii 2/255 and 4/55 and evaluate robustness on all datasets.

**VLM interpretation.** The visualization in Fig. 7 is implemented by VLM-Visualizer[3].

## A.4 ADDITIONAL RESULTS

**Quantitative Evaluation on VLMs Interpretations.** In this section, we provide a quantitative assessment of how AFT of CLIP enhances the visual interpretability of downstream VLMs. Since there are currently no established metrics for evaluating VLM interpretations, we propose a straightforward evaluation scheme, as illustrated in Fig. 8. We selected test images from the Imagenette dataset, which consists of 10 easily classified categories from ImageNet. We then prompted LLaVa with the following request:

*Please select one class from tench, English springer, cassette player, chain saw, church, French horn, garbage truck, gas pump, golf ball, parachute to describe the image. Only output the class name.*

Typically, LLaVa responds with just the class name, such as "Church," which is generally made up of 1 to 3 tokens. Including the initial colon and the end-of-sentence tokens, we obtain a few tokens that correlate with several attention maps. We compute various metrics for each attention map using the pointing game metric, which encompasses PG, PG-energy, pixel accuracy (Pixel Acc.), average precision (AP), and average mask intersection over union (maskIoU). Additionally, we calculate the Gini Index (Hurley & Rickard, 2009) to measure the sparsity of the attention map. For each case, we take the highest metric value across different tokens. Since the pointing game

---

[1]https://github.com/LAION-AI/CLIP_benchmark
[2]https://github.com/mlfoundations/open_clip
[3]https://github.com/zjysteven/VLM-Visualizer

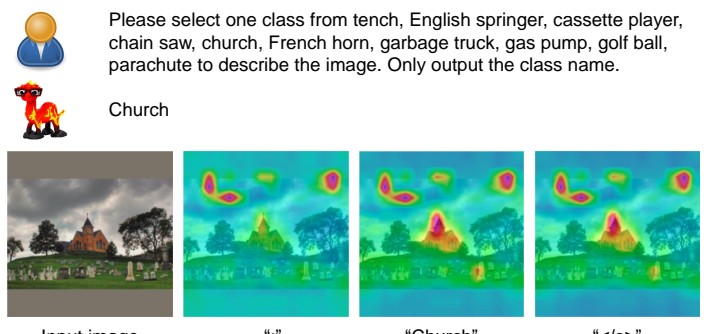

Please select one class from tench, English springer, cassette player, chain saw, church, French horn, garbage truck, gas pump, golf ball, parachute to describe the image. Only output the class name.

Church

| Input image | ":" | "Church" | "" |

Figure 8: Illustration on how we utilize LLaVa to perform image classification task and generate corresponding attention map for each response token.

Table 3: Quantitative evaluation of attention maps generated by LLaVa. We measure the Gini Index (Gini), Point Game (PG and PG-energy) and Segmentation test (AP and MaskIoU) on the interaction between Imagenette and ImageNet-Segmentation validation dataset.

| Encoder | Gini↑ | PG↑ | PG-energy↑ | Pixel Acc.↑ | AP↑ | mask-IoU↑ |
|---|---|---|---|---|---|---|
| CLIP | 25.29 | 5.56 | 19.71 | 69.62 | 28.77 | 4.13 |
| CLIP w/ AFT | **29.16** | **7.64** | **21.82** | **72.14** | **33.30** | **4.78** |

requires a segmentation mask, we limited our analysis to images that overlap between Imagenette and ImageNet-Segmentation, resulting in a total of 144 images.

We compare the outcomes of LLaVa using the CLIP encoder against those using the CLIP encoder with AFT. The results, presented in Table 8, show that employing the CLIP encoder with AFT leads to sparser and more interpretable attention maps generated by LLaVa. However, it is important to note that the current metrics have limitations. For instance, ViT attention maps frequently contain numerous high-norm tokens in low-informative background areas, a well-documented issue (Darcet et al., 2023) that AFT does not fully mitigate. These high-norm tokens significantly impact the proposed metrics. We view this work as a preliminary exploration and aim to identify better metrics for a more rigorous quantitative evaluation.

**Comparison between Supervised AFT and Unsupervised AFT.** To empirically examine the differences between optimizing Eq.1 and Eq.7, we conducted experiments comparing supervised and unsupervised fine-tuning. We applied AFT to train the models using data from the COCO 2017 dataset. For supervised AFT, we utilized image captions as the $T_{\mathbf{x}}$ in Eq. 1. In contrast, for unsupervised AFT, we maintained the same configuration outlined in the main text, using CLIP ViT-L-14 as the backbone. For both models, we assessed: 1) The visual quality of the saliency maps (Fig. 9), 2) Quantitative evaluations via pointing games (Table. 4), and 3) Zero-shot accuracy (Table. 5). The results indicate that supervised AFT provides superior interpretability, particularly for Simple Gradients. This is expected, as training is steered by text descriptions, allowing the network to effectively create sparse saliency maps that align with the image content. However, the generalizability of models after supervised AFT is heavily dependent on the diversity and quality of the text descriptions. With limited training data, supervised AFT can significantly impair the model's zero-shot capabilities. Conversely, unsupervised AFT tends to achieve a better balance between interpretability and generalizability, even when trained on smaller datasets.

**Extension to Image Encoders Other than CLIP.** As noted in the main text, the proposed AFT method is versatile and can be applied to any multi-modal models that utilize embeddings to connect different modalities. It can also be extended to representation learning within single modalities. To demonstrate this flexibility, we conducted experiments with three models other than CLIP: 1) **ViT-B** trained by supervised learning on Imagenet, 2) **MONET** (Kim et al., 2024), which is an image-text foundation model trained on 105,550 dermatological images paired with natural language descriptions from a large collection of medical literature. It utilizes ViT-L-14 as its backbone and is trained

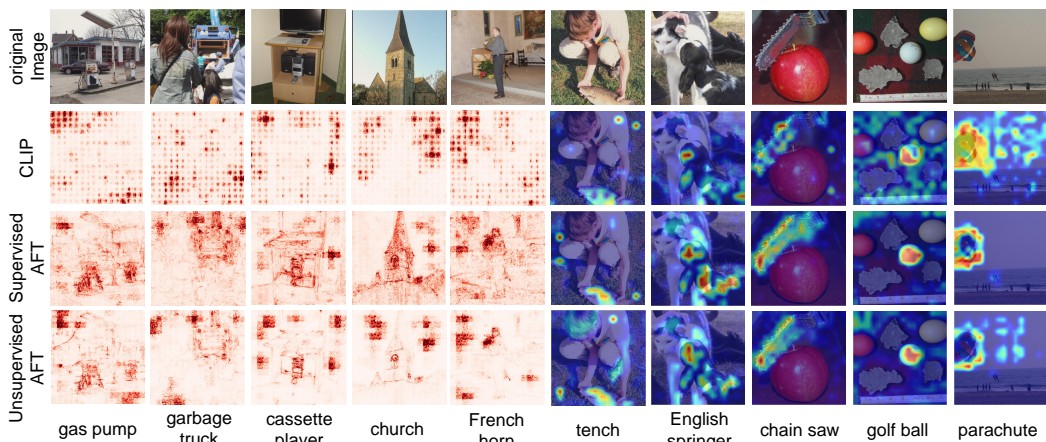

Figure 9: Comparison between supervised AFT and unsupervised AFT. The models use ViT-L-14 as the backbone and are trained on MS COCO 2017 dataset (Lin et al., 2014). We visualize saliency maps generated by both simple gradient (**left**) and GradCam (**right**).

Table 4: Evaluation of localization ability between supervised AFT and unsupervised AFT using the Point Game (PG and PG-energy) and Segmentation test (AP and MaskIoU). Evaluation performed on Simple Gradients, Grad-Cam, and Grad-ECLIP.

| Saliency maps | CLIP | PG↑ | PG-energy↑ | AP↑ | mask-IoU↑ |
|---|---|---|---|---|---|
| Simple Gradients | unsupervised | 28.88 | 31.90 | 38.14 | 2.20 |
| | supervised | **49.10** | **41.09** | **47.54** | **5.32** |
| Grad-Cam | unsupervised | **60.41** | **59.02** | **59.66** | **16.49** |
| | supervised | 58.02 | 57.86 | 59.02 | 15.79 |
| Grad-ECLIP | unsupervised | 86.44 | **61.75** | 75.67 | 22.90 |
| | supervised | **87.55** | 60.40 | **77.24** | **25.46** |

with contrastive learning, 3) **GLoRIA** (Huang et al., 2021), an attention-based model designed to learn both global and local representations by contrasting sub-regions of chest X-ray images with corresponding words in paired reports. It utilizes ResNet-50 as its backbone. For simplicity, we focused on applying AFT to the global representation of GLoRIA. We perform AFT these models with the training data from 1) Imagenet, 2) The ISIC 2024 Challenge Dataset (Kurtansky et al., 2024), and 3) MIMIC-CXR (Johnson et al., 2019) respectively. All other settings remained consistent with those outlined in the main text. The explanation results, including both simple gradients and Grad-CAM, are presented in Fig. 10. The findings indicate that AFT significantly enhances the quality of saliency maps across all three models, resulting in much sparser highlighted regions compared to the original models. This improvement leads to a more focused attention on lesions or abnormal areas. Notably, the benefits of AFT are more pronounced for vision transformers than for CNNs. Additionally, since MONET and GLoRIA are tailored for healthcare applications, these results underscore the potential of our method to support decision-making in high-stakes scenarios.

**Comparison with FARE (Schlarmann et al., 2024).** FARE is a recent work that also study adversarial training on the context of multi-modal models. While FARE aims at improving the robustness of the CLIP model to adversarial perturbations, our proposed method's primary goal is to structure the saliency maps and thus improve the CLIP's interpretability. Therefore, while both these methods result in AT-based min-max optimization problems, their different goals lead to different loss functions, where in FARE there exist $\ell_\infty$-norm hard constraints on perturbations to address norm-bounded adversarial attacks, and in our proposed AFT method, the loss function is piecewise quadratic which is the Fenchel dual to the Huber loss in our original problem formulation.

To further illustrate, we conduct experiments to compare their effects on the visual interpretability of CLIP. Specifically, we compare the ROAR on Imagenette and CUB-200-2011 (Fig. 11) and pointing

Table 5: Zero-shot accuracy evaluation on image classification datasets of CLIP model trained with supervised AFT (Sup.) and unupervised AFT (Un.).

| AFT | ImageNet | CIFAR10 | STL-10 | CIFAR100 | Cars | CalTech | OxfordPets | Flowers | DTD | EuroSAT | FGVC | PCAM | ImageNet-R | ImageNet-S | Average Zero-shot |
|---|---|---|---|---|---|---|---|---|---|---|---|---|---|---|---|
| Un. | **56.76** | **73.93** | **93.53** | **42.96** | **59.98** | **81.77** | **86.24** | **56.71** | **42.39** | 14.50 | **24.63** | 50.19 | **73.75** | **48.25** | **57.54** |
| Sup. | 45.58 | 52.85 | 81.81 | 26.58 | 54.21 | 63.01 | 82.69 | 54.17 | 33.67 | **23.04** | 18.96 | **51.85** | 57.72 | 38.30 | 48.89 |

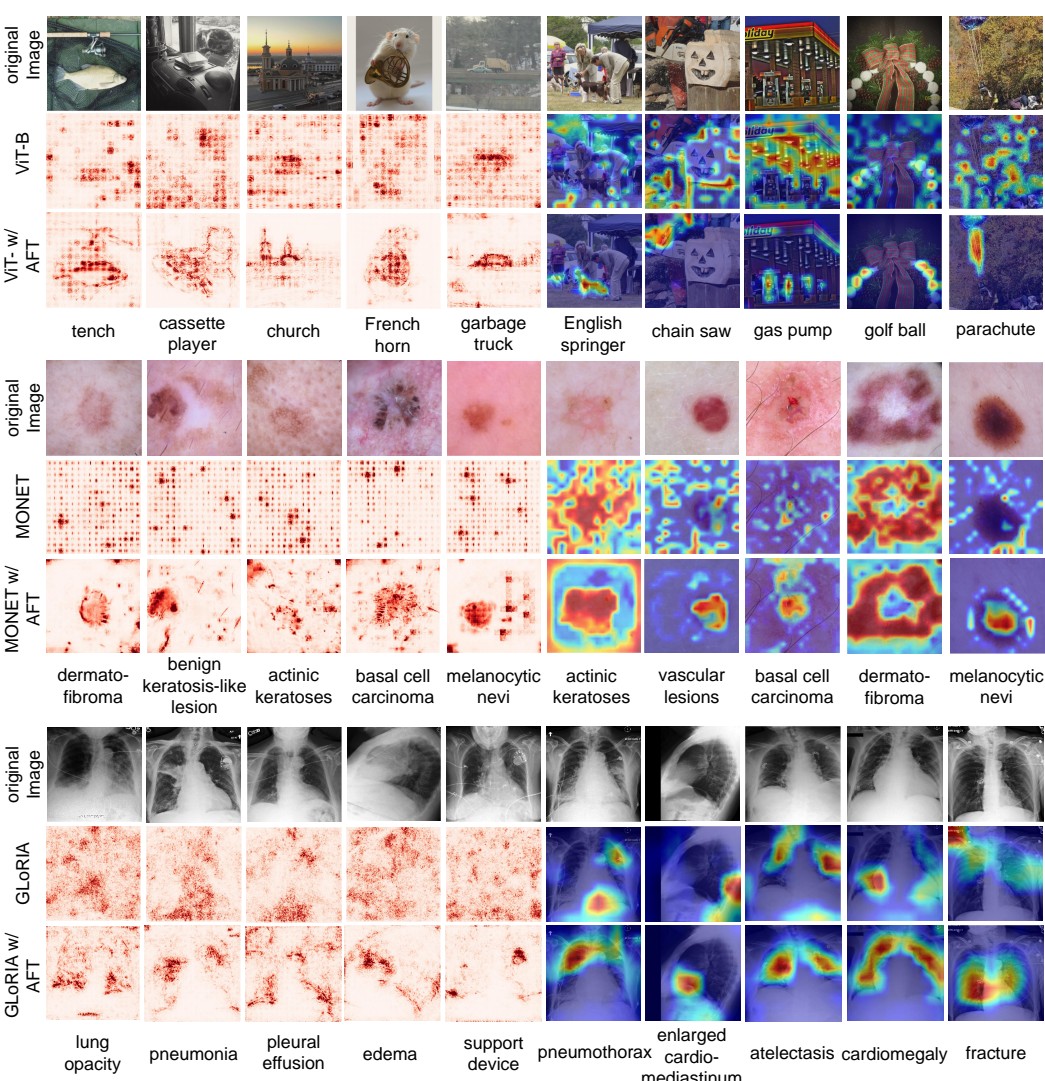

Figure 10: Extension of AFT to other image encoders. We visualize both simple gradient and GradCam. **Top**: Experimental results on supervised ViT-B. **Middle**: Experimental results on MONET (Kim et al., 2024). **Bottom**: Experimental results on GLoRIA (Huang et al., 2021).

game on ImageNet-Segmentation validation set (Table 6) between two methods. The numerical results suggest that both algorithms can improve the interpretability of the CLIP model by boosting the sparsity of saliency maps. However, there is a relative improvement of the interpretability of our proposed AFT over FARE.

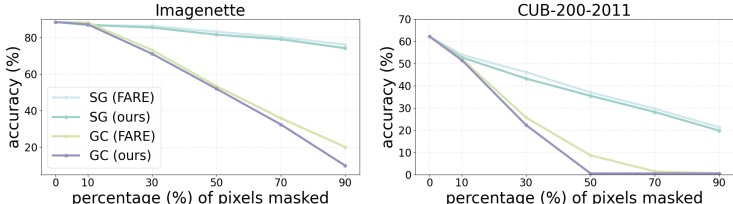

Figure 11: Comparison of Remove and Retrain (ROAR) on zero-shot classification between FARE and our method. A more rapid drop in accuracy shows the highlighted features are more important. Our algorithm design enables CLIP to catch more informative features compared with FARE.

Table 6: Comparison of localization ability between FARE and our proposed AFT. We experiment with the Point Game (PG and PG-energy) and Segmentation test (AP and MaskIoU) on the ImageNet-Segmentation validation dataset.

| Saliencymaps | CLIP | PG↑ | PG-energy↑ | AP↑ | mask-IoU↑ |
|---|---|---|---|---|---|
| SG | FARE | 30.06 | 31.68 | 38.11 | 2.15 |
|  | ours | **33.19** | **33.94** | **39.78** | **2.21** |
| GC | FARE | 70.05 | 64.75 | 66.11 | 21.50 |
|  | ours | **74.31** | **65.47** | **67.23** | **22.46** |
| LLaVa attention | FARE | 5.56 | 21.47 | 32.11 | 4.39 |
|  | ours | **7.64** | **21.82** | **33.30** | **4.78** |

**Results of Different Network Architectures.** In Fig. 12, we showcase several simple gradient maps corresponding to different network architectures, including ResNet-50, ViT-B, and ViT-L. We can see the trend as the network becomes larger and more complex, the saliency maps also become more difficult to understand. On the other hand, the proposed AFT can consistently improve the quality of the saliency maps, making the simple gradient more human-understandable.

**Results of Different Regularization.** We visualize saliency maps trained on AFT with different functions as regularization. We replace $h^*(\cdot)$ in Eq. 4 with different functions, including the smoothed version of the elastic net:

$$h^\star(\mathbf{u}) = \epsilon_1 \sum_i H_\eta(\mathbf{u}_{(i)}) + \epsilon_2 \|\mathbf{u}\|_2^2, \epsilon_1 > 0, \epsilon_2 > 0, \tag{21}$$

smoothed version of group-norm:

$$h^\star(\mathbf{u}) := \epsilon \sum_i H_\eta(\|\mathbf{u}_{S_i}\|_2), \epsilon > 0, \tag{22}$$

where $S_1, \cdots, S_t \subseteq \{1, \cdots, d\}$ are disjoint variable subsets. The visualization results are shown in Fig. 13. $L_1$-norm and elastic nets have similar effects on the Simple Gradient maps, while group-norm regularization makes the saliency map more compact and connected.

**Effects of Regularization Strength.** We change the regularization strength and visualize the resulting simple gradient maps in Fig. 14. Even with a very small $\epsilon$ such as $1/255$, the visual quality of the Simple Gradient maps has significant improvement, which will cause very small drops of zero-shot accuracy as shown in Table 2 in the main text. As we increase the $\epsilon$, we can see an obvious pattern that the saliency maps become more and more concise.

**Effects of Training iterations.** We can show how the Simple Gradient maps evolve during the fine-tuning process. We showcase the simple gradient with different fine-tuning iterations in Fig. 15. The Simple Gradient becomes more concise and cleaner as the fine-tuning processing. However, with a few fine-tuning steps, the resulting saliency map can already be of good visual quality. This shows our method is computationally efficient.

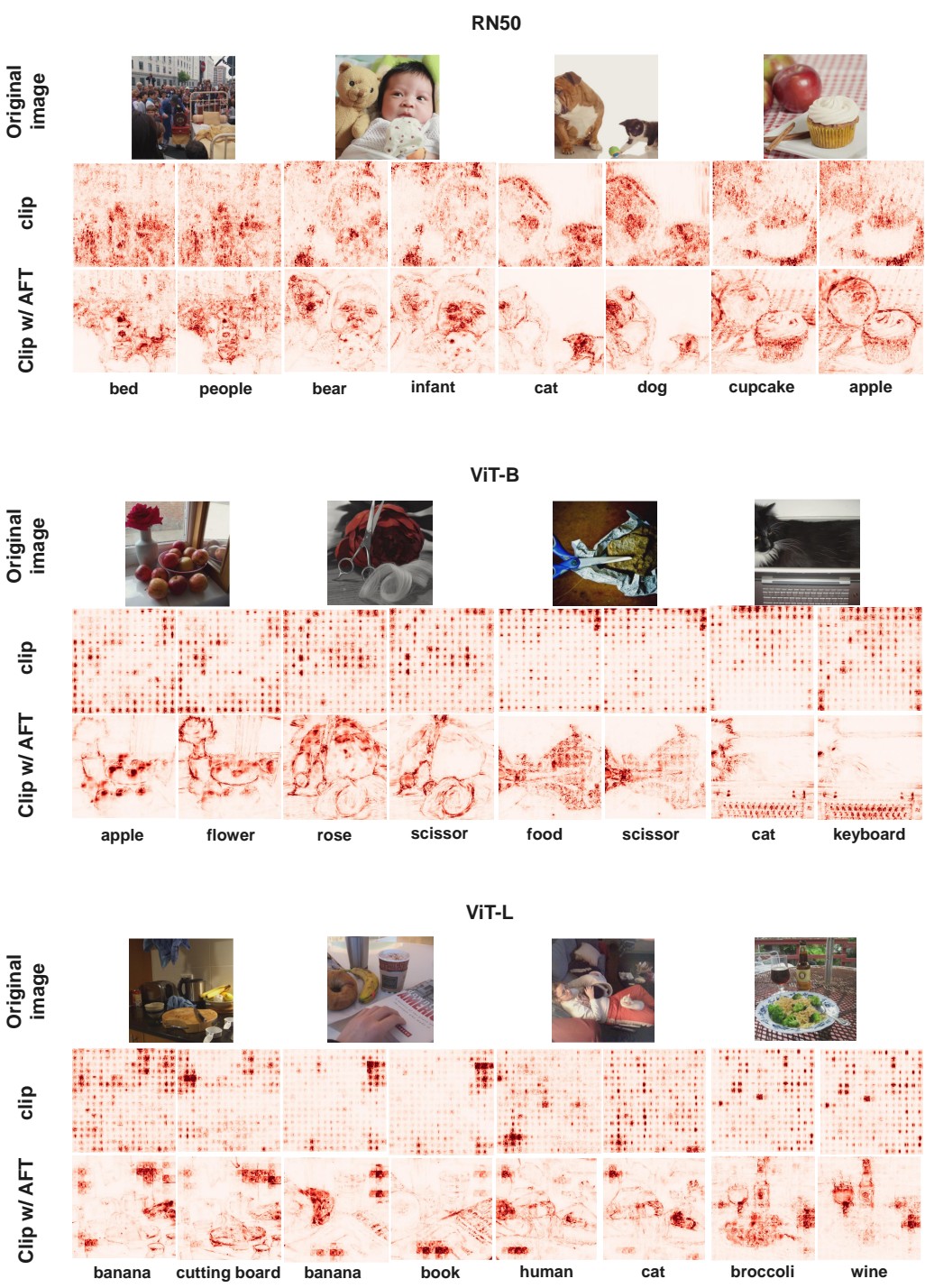

Figure 12: Comparison of Simple Gradient maps w/wo adversarial fine-tuning among different network architectures.

**Interpretation Robustness.** We further study the robustness of the saliency maps before and after AFT. As the SG of the original CLIP shows an extremely noisy pattern, we focus on GC in this study. We inject small random Gaussian noise to the input, which is imperceptible to humans. Then

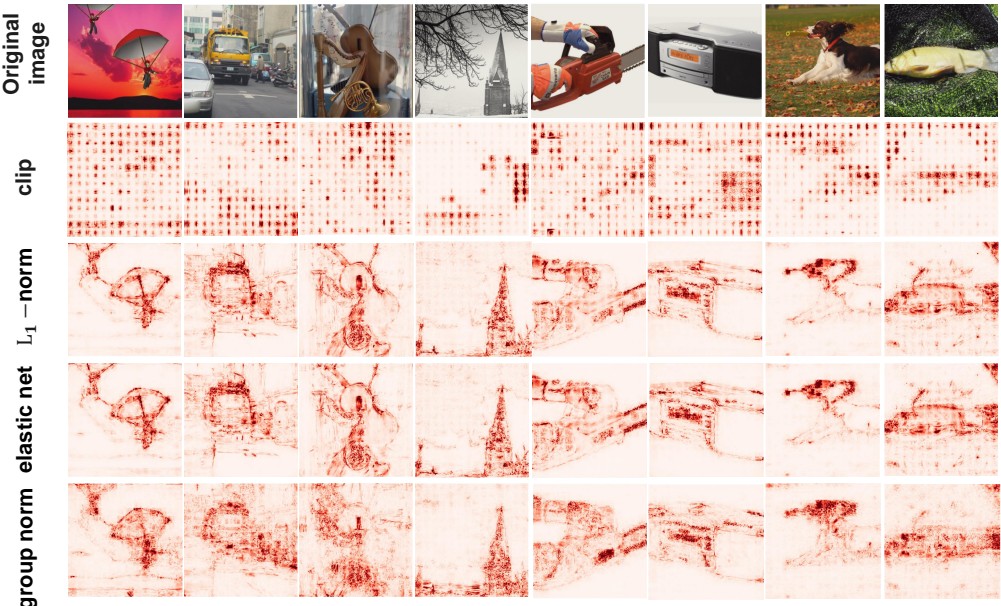

Figure 13: Comparison of Simple Gradient maps w/wo adversarial fine-tuning subject to different regularizations.

Table 7: SSIM (%) between Grad-Cam maps before and after injecting Gaussian noise with SD of $\sigma$ to the input.

| $\sigma$ | 1/255 | 3/255 | 5/255 | 7/255 | 9/255 |
|---|---|---|---|---|---|
| CLIP | 91.18 | 82.98 | 77.56 | 73.62 | 70.58 |
| CLIP W/ AFT | 99.99 | 99.96 | 99.79 | 99.37 | 98.55 |

we measure the similarity of the GC with the clean GC, represented by SSIM. The results in Table 7 reflect that the GC of the original CLIP is susceptible to random noise, with tiny noise causing great changes. On the other hand, AFT makes the interpretation more robust to perturbations, with significant improvement of SSIM over the non-AFT counterpart.

**Inter-class Similarity.** As posited by Han et al. (2023), there exists a correlation between the model interpretability and class-similarity information. An interpretable model would possess a greater understanding of inter-class similarity, thereby generating more similar predictions on images belonging to different but related classes. To assess whether this finding extends to AFT, we divided the 1,000 classes of the ImageNet dataset into 66 categories based on the coarse classification scheme proposed by Eshed (2020), excluding the 'other' category. We predict the category logit with the visual encoder via linear probing and quantify the entropy of classes within the same category, which represents the amount of information contained in the model for that category. The results in Table 8 demonstrate an improvement of class-similarity information after AFT. The smoothness imposed during the AFT can enhance the semantic-level smoothness of the prediction, which increases similarity in the representations of samples from similar classes.

**Zero-shot Accuracy and Robustness.** Here we show the complete version of Table 2 in Table 9. As a reference, we include the results of FARE (Schlarmann et al., 2024), which utilizes AFT to enhance the adversarial robustness of CLIP. AFT can result in some drop in terms of clean accuracy. But with a minor $\epsilon$ such as 1/255, the zero-shot accuracy only drops by 2.84%, while the performance on ImageNet even increases by 1.02%, while such regularization can still help improve the visual interpretability greatly, as shown in Fig. 14. Moreover, AFT can improve the robustness of the CLIP model significantly, which is another desired property for application in safety-critical industries.

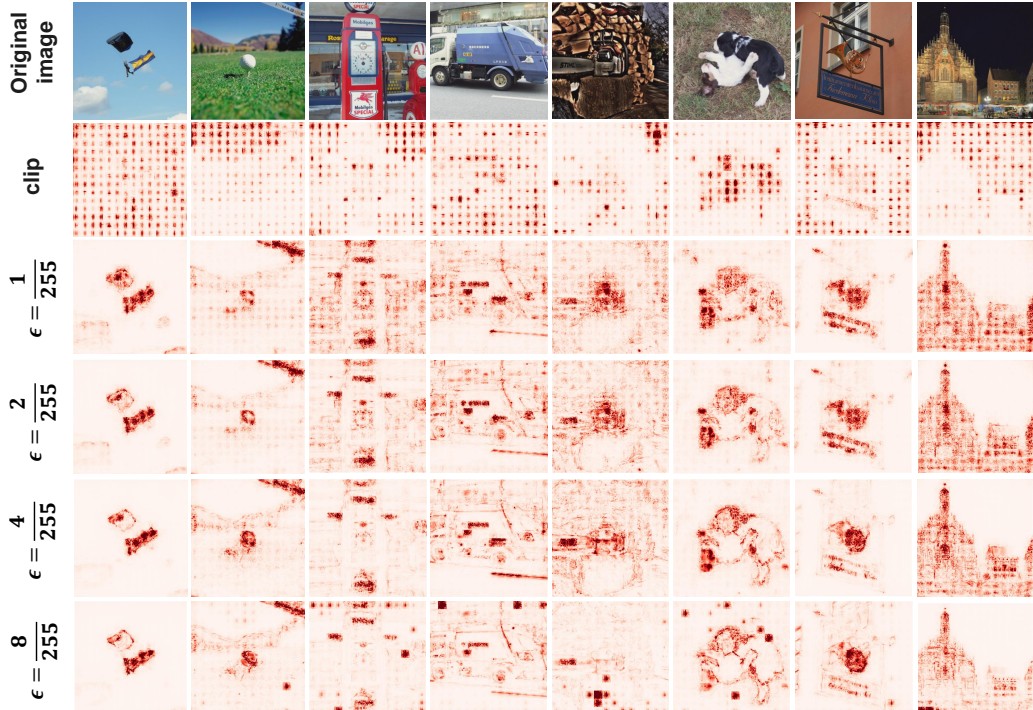

Figure 14: Comparison of Simple Gradient maps w/wo adversarial fine-tuning with different regularization strength.

Table 8: Comparison of the entropy measured based on the output of all class (Entire) and output of classes in the same category to the correct class (Category).

| Model | Entire | Category |
|---|---|---|
| supervised | 1.32 | 0.15 |
| CLIP | **5.75** | 2.06 |
| CLIP w/ AFT | 5.44 | **2.09** |

We also discovered our regularized adversarial training achieves comparable robustness compared with the state-of-the-art adversarial training method FARE.

**User Study.** We conducted a small-scale user study following the methodology of Kim et al. (2022) to quantify the impact of high-quality saliency maps on user decision-making. Our focus was on the "distinct" task, as illustrated in Fig. 16. Specifically, we collected 10 validation cases from ImageNet for both the original CLIP and CLIP with AFT, where each set included 5 cases with correct zero-shot predictions and 5 with incorrect predictions. For each case, we displayed 4 Grad-CAM maps generated by the network corresponding to the top predicted classes: for correct cases, these included the correct class, and for incorrect cases, we showed the top three wrong classes alongside the true class. Users were presented with the input image and the four explanations in random order, and we asked them to select the class they believed was correct, without revealing class names to simulate a lack of domain knowledge. The 20 cases were also shuffled randomly.

We gathered experimental results from 20 users, which are summarized in Table 10. Our findings indicate that saliency maps can aid users in their decision-making process. Across all metrics, results were significantly higher than random guesses (25%). For cases with correct predictions, we observed that enhancing the quality of saliency maps had limited benefits for user performance. This is likely because humans can automatically de-noise information in their minds, making a cleaner

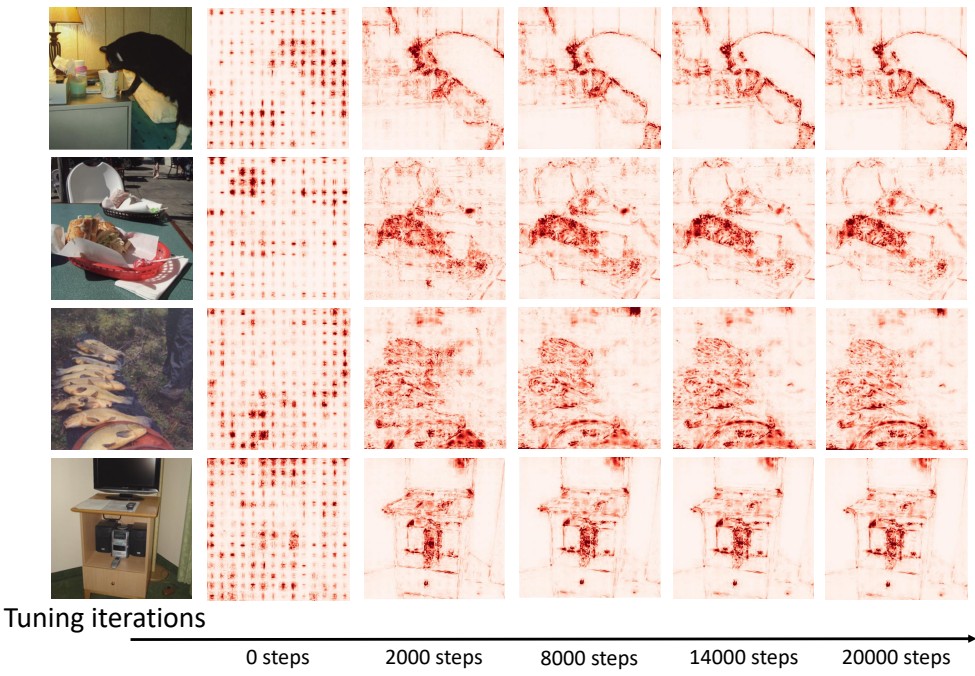

Tuning iterations

0 steps  2000 steps  8000 steps  14000 steps  20000 steps

Figure 15: Evolution of the Simple Gradient maps during the fine-tuning process.

Table 9: Accuracy evaluation on image classification datasets of CLIP model. The superscripts on the upper right side show the value of $\epsilon$ (/255) during the adversarial fine-tuning.

| $L_\infty$ | Vision Encoder | ImageNet | Zero-shot Datasets | | | | | | | | | | | | | Average Zero-shot |
| | | | CIFAR10 | STL-10 | CIFAR100 | Cars | CalTech | OxfordPets | Flowers | DTD | EuroSAT | FGVC | PCAM | ImageNet-R | ImageNet-S | |
|---|---|---|---|---|---|---|---|---|---|---|---|---|---|---|---|---|
| clean | CLIP | 74.90 | 95.20 | 99.31 | 71.08 | 77.94 | 83.30 | 93.21 | 79.17 | 55.21 | 62.65 | 31.77 | 52.00 | 87.86 | 59.59 | 72.95 |
| | FARE[1] | 75.76 | 93.86 | 99.12 | 75.27 | 74.36 | 84.62 | 92.97 | 75.65 | 54.73 | 31.35 | 28.95 | 52.44 | 87.53 | 60.51 | 70.10 |
| | ours[1] | 75.92 | 94.14 | 99.11 | 75.34 | 74.23 | 84.47 | 92.91 | 75.74 | 54.47 | 31.61 | 28.71 | 52.70 | 87.56 | 60.42 | 70.11 |
| | FARE[2] | 74.24 | 89.52 | 98.47 | 69.13 | 70.53 | 84.77 | 91.06 | 70.60 | 50.05 | 25.39 | 26.70 | 50.01 | 85.52 | 59.73 | 67.04 |
| | ours[2] | 74.60 | 89.71 | 98.50 | 69.66 | 70.15 | 85.03 | 91.09 | 70.34 | 50.00 | 24.74 | 27.57 | 50.02 | 85.45 | 59.58 | 67.06 |
| | FARE[4] | 70.40 | 77.67 | 96.04 | 56.53 | 63.84 | 84.70 | 87.14 | 58.07 | 43.83 | 18.28 | 21.99 | 50.00 | 80.20 | 56.73 | 61.16 |
| | ours[4] | 70.88 | 78.47 | 96.28 | 57.31 | 63.36 | 84.73 | 86.97 | 57.64 | 42.93 | 18.56 | 22.35 | 50.02 | 80.39 | 56.91 | 60.99 |
| $L_\infty = 2/255$ | CLIP | 0.00 | 0.00 | 0.00 | 0.00 | 0.00 | 0.00 | 0.00 | 0.00 | 0.00 | 0.00 | 0.00 | 0.00 | 0.00 | 0.00 | 0.00 |
| | FARE[1] | 29.82 | 51.80 | 83.40 | 28.10 | 13.90 | 61.80 | 52.60 | 19.20 | 19.00 | 2.00 | 2.20 | 11.30 | 41.90 | 31.00 | 32.17 |
| | ours[1] | 28.84 | 51.30 | 82.80 | 27.20 | 12.80 | 61.10 | 50.40 | 18.80 | 18.40 | 1.40 | 2.00 | 9.00 | 41.20 | 30.50 | 31.30 |
| | FARE[2] | 46.10 | 60.60 | 90.30 | 35.60 | 25.70 | 72.80 | 68.50 | 31.70 | 26.60 | 6.20 | 5.90 | 41.90 | 56.50 | 38.30 | 43.12 |
| | ours[2] | 47.36 | 61.70 | 90.80 | 37.40 | 25.50 | 73.80 | 68.60 | 31.70 | 26.50 | 8.60 | 5.90 | 46.90 | 57.40 | 38.70 | 44.12 |
| | FARE[4] | 52.44 | 57.10 | 89.50 | 36.70 | 29.80 | 76.80 | 72.50 | 31.50 | 28.30 | 12.80 | 8.20 | 50.20 | 61.60 | 41.60 | 45.89 |
| | ours[4] | 53.68 | 57.90 | 90.20 | 38.00 | 30.70 | 77.60 | 72.50 | 30.20 | 28.80 | 12.60 | 8.00 | 50.20 | 62.10 | 43.10 | 46.30 |
| $L_\infty = 4/255$ | CLIP | 0.00 | 0.00 | 0.00 | 0.00 | 0.00 | 0.00 | 0.00 | 0.00 | 0.00 | 0.00 | 0.00 | 0.00 | 0.00 | 0.00 | 0.00 |
| | FARE[1] | 2.18 | 10.50 | 29.10 | 5.90 | 0.20 | 18.10 | 3.20 | 1.10 | 3.50 | 0.00 | 0.00 | 0.30 | 12.00 | 12.80 | 7.44 |
| | ours[1] | 1.82 | 10.20 | 28.40 | 5.40 | 0.20 | 17.10 | 2.30 | 0.90 | 3.10 | 0.00 | 0.00 | 0.10 | 11.70 | 13.10 | 7.12 |
| | FARE[2] | 16.64 | 25.90 | 61.70 | 14.10 | 4.80 | 45.90 | 27.90 | 7.00 | 11.80 | 0.70 | 0.60 | 17.30 | 25.60 | 22.40 | 20.44 |
| | ours[2] | 18.34 | 26.90 | 63.40 | 14.30 | 5.40 | 47.30 | 30.50 | 6.90 | 12.50 | 1.50 | 0.50 | 19.70 | 27.00 | 23.10 | 21.46 |
| | FARE[4] | 33.48 | 34.80 | 74.30 | 20.10 | 12.80 | 64.20 | 50.70 | 12.10 | 17.40 | 11.20 | 2.60 | 50.20 | 40.40 | 30.20 | 32.38 |
| | ours[4] | 35.28 | 36.40 | 75.70 | 21.20 | 12.80 | 65.90 | 51.50 | 13.00 | 17.60 | 11.30 | 2.60 | 50.20 | 41.00 | 31.60 | 33.14 |

saliency map less impactful. However, for cases with incorrect predictions, we found that better saliency maps significantly improved user decision-making. Despite the model's wrong predictions, when provided with the true class name, it still produced visually appealing and semantically aligned saliency maps. Consequently, users were more inclined to select the true class. In contrast, when faced with noisy saliency maps, users tended to de-noise them mentally, often retaining only the largest connected component. This process could lead to the exclusion of saliency corresponding to the actual object.

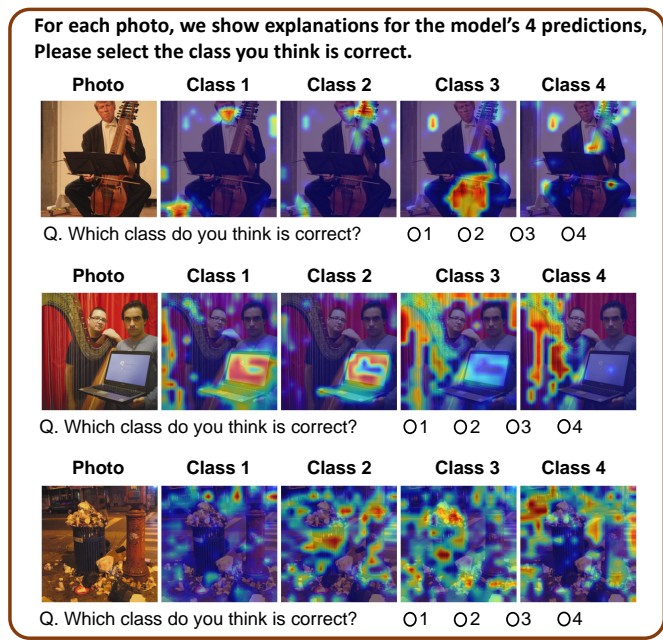

Figure 16: Illustration on our user study set ups.

Table 10: Results of user study. For each study, we report the mean accuracy and standard deviation of the participants' performance. The "correct" row refer to cases whose labels are corrected predicted by the network while "incorrect" is the opposite. Overall, explanations of higher quality somehow help the users to correct the mistakes made by the network.

| Backbone | CLIP | CLIP w/ AFT |
|---|---|---|
| correct | $73.00_{\pm 18.19}$ | $74.00_{\pm 18.00}$ |
| incorrect | $40.00_{\pm 20.00}$ | $65.00_{\pm 15.33}$ |

Another noteworthy finding was that users often selected the saliency maps with the highest sparsity as corresponding to the true class. This observation supports our motivation to promote sparsity in saliency maps.

**Cost Analysis.** Finally, we briefly address the cost associated with our algorithms. AFT is a form of fine-tuning for the visual encoder of CLIP, and it does not introduce any new parameters. All aspects of the model, apart from the specific parameter values of the visual encoder after AFT, remain unchanged from the original CLIP. Consequently, the inference cost for CLIP and downstream VLMs remains identical. To minimize the cost of AFT, we applied unsupervised AFT without utilizing the text encoder, allowing it to be implemented on relatively small-scale datasets. Specifically, we report the training time for our method in Table 11, based on training with the ImageNet dataset for 2 epochs. All experiments were conducted on NVIDIA GeForce RTX 4090 GPUs.

## A.5 DISCUSSION

One of the primary objectives of this work is to enhance the quality of saliency maps. While saliency maps are among the most widely used tools for explanations, they do have limitations. For instance, they primarily indicate the presence of specific objects but struggle to elucidate more complex contextual features, such as orientation and spatial arrangement. Consequently, our numerical experiments focused on using saliency maps to highlight non-contextual features of the (image, text) pairs, in line with much of the existing literature. However, during our experiments, we discovered

Table 11: Cost analysis. Training time is calculated as training on ImageNet training set for 2 epochs.

| Backbone | Batch size | N. of GPUs used | Training time |
|----------|-----------|-----------------|---------------|
| RN50 | 128 | 2 | 20.48h |
| ViT-B-16 | 128 | 2 | 34.20h |
| ViT-L-14 | 64 | 4 | 94.28h |

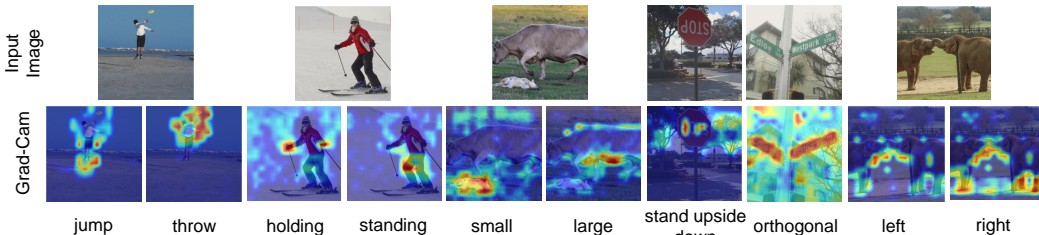

Figure 17: Combining multiple modalities in the interpretation framework enables the explanations for more complicated concepts, such as interaction, size, and relative spatial arrangement. It is still imperfect. For example, the CLIP explanation fails to distinguish between "left" and "right".

that integrating both text and image modalities within our CLIP-based interpretation framework can improve interpretative capabilities. As illustrated in Fig. 17, the combination of text prompts and saliency maps allows the framework to explain more complex concepts, such as interactions between human and objects, relative sizes, and spatial relationships. There are, of course, instances where the explanations fall short; for example, the model may struggle to differentiate between "left" and "right." Nevertheless, these findings underscore the significance of pursuing research directions focused on multi-modal interpretations.

As for the application of a more interpretable CLIP, it is widely acknowledged CLIP can be a zero-shot classifier for many tasks (Guo et al., 2023; Novack et al., 2023; Saha et al., 2024). When treating CLIP as a zero-shot classifier, it can be important to also provide the sample-level explanation as well as mechanistic interpretations to help humans make the final decision, especially in high-stake scenarios such as computer-aided diagnosis (Liu et al., 2023). Additionally, the explanations can do more. There is work utilizing the saliency maps generated by CLIP for visual grounding or open-vocabulary segmentation (Hsia et al., 2022; Lin et al., 2023). For such applications, explanations with better quality can improve the model performance. We also give preliminary evidence through the pointing game experiments. More interestingly, there is a recent study (Yu et al., 2024) that employs CLIP to generate attention prompts. The attention prompt can point out the correct region to focus on for the downstream LLM, which has proven empirically to significantly improve the performance of the large vision-language models. It is foreseeable that better localization enables the large model to better focus on the correct objects. What is more, in our main text, we also provide an example that the improved visual interpretability helps explain why the model makes certain hallucinations. Overall, it is worth the efforts to build a more interpretable CLIP.

