# OpenReview forum: "Boosting the visual interpretability of CLIP via adversarial fine-tuning"
_ICLR.cc/2025/Conference — ICLR 2025 Poster_

### Official Review · Reviewer_pAYu · 2024-10-28

**Soundness:** 3
**Presentation:** 3
**Contribution:** 3
**Rating:** 8
**Confidence:** 3

**Summary:**

In this paper, the authors propose an unsupervised adversarial fine-tuning (AFT) method with norm-regularization to enhance the visual interpretability of CLIP.  Overall, this paper proposes a comprehensive theoretical analysis of their motivation and provides extensive experiments to prove the correctness of their ideas. However, it still has some issues.

**Strengths:**

1. The theoretical analysis of their method is supportive
2. The experimental results are enough to support their motivation and demonstrate its correctness.

**Weaknesses:**

1. Too few content is with regard to the implemental details.  At least, an overview of the proposed method should be provided.

2. Though the proposed method is interesting, a single novel norm-regularization might not be enough to support the high standard of ICLR. More introduction content of your main contributions should be added. For instance, the innovation of the proposed method or the creative construction of your method.

**Questions:**

1. Could you please provide more easy-understanding and plain descriptions of what you do? For example, the architecture of your propose method, and the final loss function of your method.

2. Make more claims on your contributions. For instance, the innovation of your ideas, or the creative construction of your method.

3. why do you call your methods as the adversarial fine-tuning?

---

> ### Author Response · Authors · 2024-11-21
> **Authors’ Response to Reviewer pAYu**
>
> We thank Reviewer pAYu for his/her time and thoughtful feedback on our work. Regarding the raised comments and questions,
>
> ---
> >**Overview of the method**
>
> Thank you for the suggestions. We have included a paragraph at the beginning of the method section (line 156) as an overview of our method, and the implementation detail is in the *Appendix A.3* (line 930).
>
> As illustrated in Figure 1 of the main paper, we introduce a norm-regularized adversarial learning scheme to fine-tune the visual encoder of CLIP, enhancing its interpretability. With a pre-trained CLIP model using any backbone, we fine-tune its visual encoder on a relatively small-scale, image-only dataset using the objectives Eq. 7 in the paper.
>
> The $h(\cdot)$ in the objective is a flexible regularization term tailored to enforce specific properties in the saliency maps. Here, we mainly focus on the duality of Huber loss in the main text to enforce sparsity. We have observed that this adversarial fine-tuning approach not only improves the visual quality of the saliency maps generated by the network but also has a positive effect on the alignment of its neurons with human-understandable concepts.
>
> ---
> >**Contribution of our work**
>
> Our main contributions can be summarized in three key areas:
>
> - **Methodology**: We introduce a general adversarial training framework designed to enhance the visual interpretability of CLIP. As proposed for multi-modality models, the fine-tuning of CLIP follows a structured loss function that differs from standard classification loss.
>
> - **Duality Framework**: The duality framework presented in our paper is essential for designing the adversarial training loss function to structure saliency maps and improve the interpretability of CLIP. As discussed in Theorem 1, this framework suggests the application of the Huber loss, which results in the piecewise quadratic function in Equation (5) for performing adversarial fine-tuning (AFT). The Huber loss approximates the $L_1$-norm function, promoting sparsity, and also leads to a strongly-convex dual function to be optimized in the inner optimization of AFT.
>
> - **Analysis of transferability of the interpretation maps for multi-modal models**: We would like to note that, to the best of our knowledge, our work is the first to study whether the interpretation effect can be transferred to out-of-distribution dataset and downstream tasks.
>
> ---
> >**Regarding the method’s title**
>
> In our approach, we initialize the model parameters using the pre-trained CLIP model. The fine-tuning process aims to enhance the interpretability of the encoder while maintaining close alignment with the pre-trained weights. This alignment is crucial to ensure that the fine-tuned encoder remains compatible with the original text encoder. Additionally, our proposed objective function incorporates a two-stage minimax optimization process. This process involves adversarial elements, where the model is trained to optimize against adversarial examples or conditions. This adversarial framework is a fundamental aspect of our method, which is why we refer to it as adversarial fine-tuning.

---

> > ### Comment · Reviewer_pAYu · 2024-11-22
> >
> > Thanks for your reply! It addresses most of my concerns, thus I tend to accept it and raise my scores.

---

> > > ### Author Response · Authors · 2024-11-26
> > > **Thank you for the reply**
> > >
> > > Many thanks for your response and kind consideration. We will include your valuable suggestions in the final version of the paper.

---

### Official Review · Reviewer_cNDm · 2024-11-02

**Soundness:** 3
**Presentation:** 3
**Contribution:** 2
**Rating:** 8
**Confidence:** 4

**Summary:**

The paper proposes an unsupervised adversarial fine-tuning (AFT) method to alleviate the issues of insufficient representation and poor interpretability in visual representation learning of CLIP. The paper provides theoretical evidence for the feasibility of the method and presents a rigorous derivation process with well-defined formulas. The experimental results provided in the paper demonstrate the effectiveness of the proposed approach.

**Strengths:**

1. There is a clear definition of the research problem, which is to enhance the visual interpretability of CLIP. The problem is visualized through the form of saliency maps.
2. The proposed method in the paper demonstrates a good level of originality. The extensive formalized equations provide evidence of the method's rigor and credibility. The abundance of experimental data and visualization results further confirm the effectiveness of the AFT method.
3. The paper presents a clear and well-structured expression, making it easy to read and follow.

**Weaknesses:**

1. From the experimental results, AFT appears to be an effective method. However, it is unclear whether this method introduces additional complexity. I would like to see comparative experiments that evaluate the parameter quantity, computational cost, and other factors before and after applying this method.
2. Although the paper designs the AFT method based on CLIP and achieves good results on relevant datasets, it is mentioned that the AFT method can be extended to any base model that includes intermediate embedding layer connection patterns. More experiments should be provided for this claim. In other words, selecting a few base models other than CLIP and applying the AFT method to demonstrate its general applicability would be beneficial.

**Questions:**

1. Does AFT introduce additional complexity? It would be better to provide comparative experiments regarding the parameter number, computational cost, and other factors before and after applying this method.
2. Can the proposed method be applied to base models other than CLIP? These experiments could show the generalizability of the method.

---

> ### Author Response · Authors · 2024-11-21
> **Authors’ Response to Reviewer cNDm**
>
> We thank Reviewer cNDm for his/her time and thoughtful feedback on our work. Regarding the raised comments and questions,
>
> ---
> >**Cost analysis**
>
> AFT is a fine-tuning method for the visual encoder of CLIP that does not introduce new parameters. All aspects of the visual encoder, except for specific parameter values, remain unchanged from the original CLIP, ensuring that the inference cost for CLIP and downstream VLMs stays the same. We minimize the AFT cost by using unsupervised AFT without the text encoder, and it can be applied to relatively small-scale datasets. Specifically, we trained our method on the ImageNet training set for 2 epochs using NVIDIA GeForce RTX 4090 GPUs. The cost analysis is included in the revised manuscript, detailed in *Appendix A.4. Cost Analysis* (line 1443).
>
> |backbone|batch size|# of GPUs used|Training times|
> |-----|--------|-----|--------|
> |RN50|128|2|20.48h|
> |ViT-B-16|128|2|34.20h|
> |ViT-L-4|64|4|94.28h|
>
> ---
> >**Benefits for other vision encoders**
>
> The proposed AFT method is flexible and applicable to any multi-modal models using embeddings to link modalities, and can also be generalized to single-modality representation learning. We demonstrate this with three experiments on different models: 1) supervised ViT-B trained on ImageNet, 2) MONET  [[Kim et al., Nature Medicine 2024](https://www.nature.com/articles/s41591-024-02887-x)], an image-text foundation model for dermatological images, and 3) GLoRIA [[Huang et al., ICCV 2021](https://openaccess.thecvf.com/content/ICCV2021/html/Huang_GLoRIA_A_Multimodal_Global-Local_Representation_Learning_Framework_for_Label-Efficient_Medical_ICCV_2021_paper.html)], an attention-based model for contrasting chest x-ray image sub-regions with report words. Implementation details and results are in *Appendix A.4. Extension to Image Encoders Other than CLIP* (line 1020). The results show AFT significantly improves saliency map quality, making highlighted regions sparser and more focused on lesions or abnormalities. Vision transformers benefit more than CNNs. Additionally, results of the domain-specific models such as MONET and GLoRIA indicate our method's potential for supporting decision-making in high-stakes healthcare applications.
>
> ---
> **Reference**
> - Chanwoo Kim, Soham U. Gadgil, Alex J. DeGrave, Jesutofunmi A. Omiye, Zhuo Ran Cai, Roxana Daneshjou, Su-In Lee. "Transparent medical image AI via an image–text foundation model grounded in medical literature." Nature Medicine (2024).
> - Shih-Cheng Huang, Liyue Shen, Matthew P. Lungren, Serena Yeung. "GLoRIA: A Multimodal Global-Local Representation Learning Framework for Label-Efficient Medical Image Recognition." ICCV (2021).

---

> > ### Comment · Reviewer_cNDm · 2024-11-26
> > **Reply**
> >
> > After reading the response, I believe that my concerns have been addressed. I will raise my score.

---

> > > ### Author Response · Authors · 2024-11-26
> > > **Thank you for the reply**
> > >
> > > Many thanks for your response and kind consideration. We will incorporate the new results and discussions in the final version of the paper.

---

### Official Review · Reviewer_Wa4H · 2024-11-02

**Soundness:** 3
**Presentation:** 2
**Contribution:** 2
**Rating:** 6
**Confidence:** 3

**Summary:**

The paper studies the interpretability of VLMs like CLIP and and LVLMs like LLAVa. The authors propose a adversarial fine-tuning based loss which includes a regularization term and addition of Gaussian noise in the standard adversarial training loss. The resulting method AFT, achieves better interpretability in terms of saliency maps, feature importance etc in comparison to standard ERM based CLIP.

**Strengths:**

- The paper is nicely motivated and easy to follow.
- The set of experiments is extensive and helps to highlight the effectiveness of the proposed AFT loss.
- The experiment cover both CLIP and LVLMs like LLaVA.
- AFT yields better interpretable saliency maps and also attains zero-shot robustness similar to baselines.

**Weaknesses:**

- The novelty aspect of this work is not enough, it is known adversarially robust models yield more interpretable models [1, 2, 3], and this work also comes to a similar conclusion albeit with a different formulation of objective. Would FARE loss not yield similar interpretability?
see next points.
- The loss equation. 7 is similar to the objective of FARE [4] (upto Gaussian noise and the regularizer, h(.)) - there needs to be a discussion regarding this (which is missing in the current version - only a small sentence regarding this is found in the appendix).
- If one removes the noise term and h(.) - one retrieves FARE objective - How does this loss perform in the same setup as the figures 2 and 3?
- Would FARE not be an apt baseline in addition to original CLIP?

- Gaussian term is motivated by "avoid being stuck in the non-trivial stationary point" - using PGD for AT one can start at a random location in the $\ell_p$-ball, this randoms start already helps with not being stuck. It seems gaussian noise is sampled every iteration, as this seems an important part of the loss - the effect this term has is not discussed

[1] Etmann, C., Lunz, S., Maass, P., & Schönlieb, C. B.  On the Connection Between Adversarial Robustness and Saliency Map Interpretability. ICML 2019.

[2] Wang, Zifan, Matt Fredrikson, and Anupam Datta. Robust Models Are More Interpretable Because Attributions Look Normal. ICML 2022.

[3] Ross, A., & Doshi-Velez, F. (2018). Improving the Adversarial Robustness and Interpretability of Deep Neural Networks by Regularizing Their Input Gradients. Proceedings of the AAAI Conference on Artificial Intelligence, 32(1).

[4] Schlarmann, C., Singh, N. D., Croce, F., & Hein, M. Robust CLIP: Unsupervised Adversarial Fine-Tuning of Vision Embeddings for Robust Large Vision-Language Models. ICML 2024.

**Questions:**

Some questions are already present in Weakness section. A few general queries are listed here.

- The experiment with LLaVA is interesting, how much does the sparsity inducing term h(.) help here? It is interesting as now attention weights from LLM are used and this should have an effect in noise/smoothing issues mentioned earlier in the paper regarding h(.).

- Is the gaussian noise sampled at every PGD iteration?

---

> ### Author Response · Authors · 2024-11-21
> **Authors’ Response to Reviewer Wa4H**
>
> We thank Reviewer Wa4H for his/her time and thoughtful feedback on our work. Regarding the raised comments and questions,
>
> ---
> >**Novelty of our proposed method**
>
> We thank the reviewer for pointing out the references on the connection between AT and interpretation of neural network classifiers. We have discussed the references in the revised introduction. Please note that the existing works on connecting interpretability and adversarial training focus on a standard uni-modal (image modality) neural net classifier trained using robust AT **classification** loss. On the other hand, our work studies the interpretability of the **multi-modal** CLIP model (with both text and image modalities), whose training and fine-tuning follows a differently-structured loss function than the standard classification problem.
>
> Furthermore, the duality framework used in our paper can be used to properly design the adversarial training loss function for structuring the saliency maps and improving the interpretability of CLIP. As we discussed in Theorem 1, the duality framework hints at the application of Huber loss, which leads to the piecewise quadratic function in Equation (5) for performing AFT. Notably, the Huber loss mimics the $L_1$-norm function which promotes sparsity, and in addition leads to a strongly-convex dual function to be optimized in the AFT inner optimization. Therefore, we believe that the choice of the Huber loss is aligned with the goal of boosting the sparsity and interpretability of the saliency map.
>
> Finally, we would like to note that, to the best of our knowledge, our work is the first to study whether the interpretation effect can be transferred to out-of-distribution dataset and downstream tasks. Moreover, we do not know any previous work that, as in our work, extends the exploration from feature attribution to neuron concept-based interpretation, which has the potential to benefit the trending research of mechanistic interpretation today, especially for multi-modal models.
>
> ---
> >**Connections between our work and the FARE method**
>
> As discussed in the text, FARE [[Schlarmann et al., ICML 2024](https://arxiv.org/abs/2402.12336)] and our method are both applying adversarial training to the training of CLIP mode. The main difference between the two algorithms is about their target goal: FARE aims at improving the robustness of the CLIP model to adversarial perturbations, whereas our proposed method’s main goal is to structure the saliency maps and thus improve the CLIP’s interpretability. Therefore, while both these methods result in AT-based min-max optimization problems, their different goals lead to different loss functions, where in FARE there exist $\ell_\infty$-norm hard constraints on perturbations to address norm-bounded adversarial attacks, and in our proposed AFT method, the loss function is piecewise quadratic which is the Fenchel dual to the Huber loss in our original problem formulation. We have included the discussion in the introduction (line 88) of the revised paper.

---

> ### Author Response · Authors · 2024-11-21
> **Authors’ Response to Reviewer Wa4H (cont.)**
>
> ---
> >**Numerical Comparison between our AFT and FARE**
>
> In the revised introduction we have clarified the connection and different goals of the FARE algorithm and our proposed method. In addition, we performed several experiments on evaluating the effect of FARE and proposed AFT on interpretability of the CLIP model. Our numerical results indicate that, compared to vanilla CLIP, both algorithms can improve the visual interpretability of the CLIP model. Measuring scores of interpretability, we found an improvement of the interpretability scores of AFT over FARE, for both ROAR experiments and pointing games as listed in the following table (mean $\pm$ std). Details are in the *Appendix A.4 Comparison with FARE* (line 1071).
>
> - Pointing game comparison between FARE and our AFT.
>
> |Saliency Maps|method|PG|PG-energy|AP|maskIoU|
> |-----|--------|-----|--------|--------|-----|
> |Simple Gradients|FARE|30.06$_{\pm0.46}$|31.68$_{\pm0.34}$|38.11$_{\pm0.29}$|2.15$_{\pm0.28}$|
> |Simple Gradients|ours|**33.19**$_{\pm0.42}$|**33.94**$_{\pm0.37}$|**39.78**$_{\pm0.25}$|**2.21**$_{\pm0.26}$|
> |Grad-Cam|FARE|70.05$_{\pm0.41}$|64.75$_{\pm0.13}$|66.11$_{\pm0.12}$|21.50$_{\pm0.19}$|
> |Grad-Cam|ours|**74.31**$_{\pm0.39}$|**65.47**$_{\pm0.11}$|**67.23**$_{\pm0.09}$|**22.46**$_{\pm0.22}$|
> |LLaVa attention|FARE|5.56$_{\pm0.23}$|21.47$_{\pm0.08}$|32.11$_{\pm0.14}$|4.39$_{\pm0.08}$|
> |LLaVa attention|ours|**7.64**$_{\pm0.19}$|**21.82**$_{\pm0.07}$|**33.30**$_{\pm0.09}$|**4.78**$_{\pm0.06}$|
>
> - Remove and Retrain analysis on Grad Cam. Tested accuracy after x\% of pixels masked are reported.
>
> |dataset|method|10%|30%| 50% | 70%| 90%|
> |-----|--------|-----|--------|--------|-----|-----|
> |Imagenette|FARE|88.15$_{\pm0.15}$|73.07$_{\pm0.10}$|53.30$_{\pm0.15}$|35.85$_{\pm0.17}$|20.05$_{\pm0.14}$|
> |Imagenette|ours|**87.13**$_{\pm0.18}$|**71.03**$_{\pm0.16}$|**51.89**$_{\pm0.10}$|**32.39**$_{\pm0.10}$|**9.86**$_{\pm0.03}$|
> |CUB-200-2011|FARE|52.04$_{\pm0.18}$|25.66$_{\pm0.13}$|8.66$_{\pm0.17}$|1.42$_{\pm0.03}$|0.52$_{\pm0.01}$|
> |CUB-200-2011|ours|**51.56**$_{\pm0.13}$|**22.26**$_{\pm0.12}$|**0.52**$_{\pm0.01}$|**0.52**$_{\pm0.01}$|**0.52**$_{\pm0.01}$|
>
> ---
> >**Importance of Gaussian Smoothing in our designed algorithm**
>
> Note that the general AFT min-max optimization problem contains two layers of optimization: minimization over the CLIP weights and maximization of the perturbation as the dual function to the Huber loss of the CLIP’s gradient. Unlike standard AT with a hard norm constraint, the AFT norm regularization is in general a soft penalty term. Therefore, it is crucial to ensure the inner maximization problem has monotone gradients to avoid any divergence of gradient ascent updates in solving the inner maximization.
>
> The Gaussian smoothing in the proposed objective function ensures that the sum of the Gaussian smoothed loss and the negative of strongly-convex regularization penalty (dual norm to Huber loss) will be a concave function with a unique maximizer and therefore guarantees the convergence of gradient updates in solving the inner maximization. Of course, the outer minimization task over CLIP weights still remains a challenging non-convex optimization; however, using the Gaussian smoothed loss function and the dual function to Huber loss, we are able to solve the inner optimization problem and accurately compute the gradients of the primal optimization problem with the Huber loss penalty.
>
> ---
> >**Clarification on LLaVa experiment**
>
> The sparsity-inducing term encourages the input gradient to be sparse. While it does not directly enforce sparsity in the attention weights, it indirectly influences them by making the network's predictions rely on only a few tokens. The LLaVa saliency map is derived by multiplying the LLM attention weights with the ViT attention weights. On one hand, the ViT attention weights become sparse after applying AFT. On the other hand, when calculating the attention weights from LLM to image tokens, the language token can be considered as $T_x$, which assigns weights to only a few image tokens.
>
> ---
> >**Is the gaussian noise sampled at every PGD iteration?**
>
> We draw the Gaussian noise vectors independently for each training sample at each PGD iteration, which is consistent with the expectation and maximization order in Equation (7). We have clarified this point in the revision.
>
> ---
> **Reference**
> - Christian Schlarmann, Naman Deep Singh, Francesco Croce, Matthias Hein. "Robust CLIP: Unsupervised Adversarial Fine-Tuning of Vision Embeddings for Robust Large Vision-Language Models." ICML (2024).

---

> > ### Comment · Reviewer_Wa4H · 2024-11-22
> >
> > Thanks for addressing my concerns.
> > I am satisfied - It would be good if the authors would add the new comparisons to the polished version of the paper, this adds an additional baseline and improves the quality of paper.
> > I have raised my score.

---

> > > ### Author Response · Authors · 2024-11-26
> > > **Thank you for the reply**
> > >
> > > Many thanks for your response and kind consideration. We will incorporate the new results and discussions in the final version of the paper.

---

### Official Review · Reviewer_jkum · 2024-11-03

**Soundness:** 3
**Presentation:** 3
**Contribution:** 3
**Rating:** 6
**Confidence:** 3

**Summary:**

This paper proposes an unsupervised adversarial fine-tuning (AFT) strategy for improving the visual
interpretability of CLIP. Understanding and improving the interpretability of CLIP is crucial since it has been utilized as a visual encoder in multiple image generation and vision-language models (VLMs). This paper demonstrates the effectiveness of the proposed method in improving interpretability, both qualitatively and quantitatively, on multiple feature attribution and neuron explainability methods.

**Strengths:**

1. The paper is well-motivated, making it easier to follow the main ideas and claims.
2. The theoretical result on the duality between the regularised norm of adversarial perturbations and the input gradients is interesting and non-trivial. Further, the proposed unsupervised objective function helps alleviate the need for paired image-text datasets.
3. Improvements in interpretability have been demonstrated on multiple feature attribution and neuron explainability methods.
4. The proposed method does not significantly change the zero-shot generalization ability of CLIP while improving interpretability, as demonstrated by the linear probing results.

**Weaknesses:**

1. One of the primary motivations of this work is to improve the interpretability and performance of VLMs. However, only qualitative results are provided in Sec 4.3 and Fig 7, and the paper does not provide any quantitative results for the same.
2. VLMs and image generation models need vision encoders without any explicit need for encoders trained with contrastive loss. Is this method applicable to vision encoders in general or limited to CLIP?
3. The empirical performance difference between training unsupervised upper bound in Eq 7 vs Eq 1 has not been quantified. $T_x$ calculation should not significantly increase the computation overhead on smaller datasets and quantifying the difference in performance would give a better understanding of the tradeoff with use of Eq 7.
4. Ln 206: Should be $T_x$

**Questions:**

Please see weaknesses [1-3]

---

> ### Author Response · Authors · 2024-11-21
> **Authors’ Response to Reviewer jkum**
>
> We thank Reviewer jkum for his/her time and thoughtful feedback on our work. Regarding the raised comments and questions,
>
> ---
> >**Quantitative evaluation on VLMs interpretability**
>
> To the best of our knowledge, there are no existing metrics to quantitatively evaluate the properties of saliency maps in VLMs. Therefore, to address the reviewer’s comment, we attempted a simple scheme to perform quantitative evaluation. We used test images from the Imagenette dataset, a subset of 10 ImageNet classes. Using the following prompt:
>
> `Please select one class from tench, English springer, cassette player, chain saw, church, French horn, garbage truck, gas pump, golf ball, parachute to describe the image. Only output the class name.`
>
> LLaVa outputs a class name, such as “Church.” We then generated attention maps for the tokens within the class names and calculated the pointing game metric and the Gini Index to measure sparsity. Details are in *Appendix A.4 Quantitative Evaluation on VLMs Interpretations* (line 954). We compare the result between LLaVa using CLIP as encoder and LLaVa using CLIP w/ AFT as encoder. The result are as follows.
>
>
> ||Gini|PG|PG-energy|Pixel Acc.|AP|maskIoU|
> |-----|--------|--------|-----|--------|--------|--------|
> |CLIP|25.29|5.56|19.71|69.62|28.77|4.13|
> |CLIP w/ AFT|**29.16**|**7.64**|**21.82**|**72.14**|**33.30**|**4.78**|
>
>
> We show quantitatively, using the CLIP encoder w/ AFT, the attention maps generated by LLaVa become more sparse and show better interpretability. We have included the results in the *Appendix A.4 Quantitative Evaluation on VLMs Interpretations* of the revised manuscript. While our current metric has limitations, such as high-norm tokens in low-informative background areas affecting calculations [[Darcet et al., ICLR 2024](https://arxiv.org/abs/2309.16588)], this can serve as a preliminary exploration.
>
> ---
> >**Benefits for other multi-modal models**
>
> We note that the developed AFT method is flexible and applicable to any multi-modal models using embeddings to link modalities, and can also be generalized to single-modality representation learning. To demonstrate this point, we ran three experiments on different models: 1) supervised ViT-B trained on ImageNet, 2) MONET [[Kim et al., Nature Medicine 2024](https://www.nature.com/articles/s41591-024-02887-x)], an image-text foundation model for dermatological images, and 3) GLoRIA [[Huang et al., ICCV 2021](https://openaccess.thecvf.com/content/ICCV2021/html/Huang_GLoRIA_A_Multimodal_Global-Local_Representation_Learning_Framework_for_Label-Efficient_Medical_ICCV_2021_paper.html)], an attention-based model for contrasting chest x-ray image sub-regions with report words. Implementation details and results are in *Appendix A.4. Extension to Image Encoders Other than CLIP* (line 1020). The results show AFT significantly improves saliency map quality, making highlighted regions sparser and more focused on lesions or abnormalities. Vision transformers benefit more than CNNs. Additionally, results of the domain-specific models such as MONET and GLoRIA indicate our method's potential for supporting decision-making in high-stakes healthcare applications.

---

> > ### Author Response · Authors · 2024-11-21
> > **Authors’ Response to Reviewer jkum (cont.)**
> >
> > ---
> > >**Comparison between supervised AFT and unsupervised AFT**
> >
> > To address this comment, we ran numerical experiments and compared supervised AFT and unsupervised AFT on the COCO 2017 dataset. For supervised AFT, we used image captions as $T_x$ in Eq 1.Implementation details are in *Appendix A. 4 Comparison between Supervised AFT and Unsupervised AFT* (line 1006). We evaluated the two models based on 1) saliency map quality, 2) pointing games, and 3) zero-shot accuracy. The results are as follows.
> >
> >
> >
> > Table 1: Pointing games comparison bewtten unsupervised AFT and supervised AFT.
> >
> > |saliency maps|AFT|PG|PG-energy|AP|maskIoU|
> > |-----|--------|--------|-----|--------|--------|
> > |Simple Gradients|unsupervised|28.88|31.90|38.14|2.20|
> > |Simple Gradients|supervised|**49.10**|**41.09**|**47.54**|**5.32**|
> > |GradCam|unsupervised|**60.41**|**59.02**|**59.66**|**16.49**|
> > |GradCam|supervised|58.02|57.86|59.02|15.79|
> > |Grad-ECLIP|unsupervised|86.44|**61.75**|75.67|22.90|
> > |Grad-ECLIP|supervised|**87.55**|60.4|**77.24**|**25.46**|
> >
> > Table 2: Average zero-shot accuracy across 14 datasets.
> >
> > |AFT|accuracy(%)|
> > |-----|--------|
> > |unsupervised|**57.54**|
> > |supervised|48.89|
> >
> > The results suggest that supervised AFT enhances interpretability, particularly for simple gradient maps, because the text descriptions guide the training, making the saliency maps more focused on relevant image content. However, the model's generalizability after supervised AFT heavily depends on the diversity and quality of the text descriptions, and limited training data can significantly impair its zero-shot capability. In contrast, unsupervised AFT achieves a better balance between interpretability and generalizability, even with small datasets.
> >
> > ---
> > >**Typo**
> >
> > We thank the reviewer for pointing out this typo. We have corrected it in the revision.
> >
> > ---
> > **Reference**
> > - Timothee Darcet, Maxime Oquab, Julien Mairal, Piotr Bojanowski. "Vision transformers need registers." ICLR (2024).
> > - Chanwoo Kim, Soham U. Gadgil, Alex J. DeGrave, Jesutofunmi A. Omiye, Zhuo Ran Cai, Roxana Daneshjou, Su-In Lee. "Transparent medical image AI via an image–text foundation model grounded in medical literature." Nature Medicine (2024).
> > - Shih-Cheng Huang, Liyue Shen, Matthew P. Lungren, Serena Yeung. "GLoRIA: A Multimodal Global-Local Representation Learning Framework for Label-Efficient Medical Image Recognition." ICCV (2021).

---

> > > ### Comment · Reviewer_jkum · 2024-11-26
> > >
> > > I thank authors for their detailed response. While most of my concerns are resolved, the following issue still remains:
> > >
> > > The proposed quantitative evaluation on VLMs is interesting but not conclusive of the proposed method's ability to improve performance of VLMs. The performance should be compared on more real-world and diverse benchmarks such as LLaVA-Bench proposed in [1].
> > >
> > > While I am positive about this work, I plan to keep my score at 6 unless above concerns are resolved.
> > >
> > > [1] Liu, Haotian, et al. "Visual instruction tuning." Advances in neural information processing systems 36 (2024).

---

> > > > ### Author Response · Authors · 2024-11-29
> > > > **Thank you for the reply**
> > > >
> > > > We thank Reviewer jkum for the feedback on our response and are glad to hear that our response could address most of the reviewer’s concerns. Regarding the remaining comment, we conducted experiments on LLaVa-bench (coco) with LLaVa 1.5-7B and the results (mean $\pm$ s.td.) are as follows:
> > > >
> > > > ||Conversation $(\uparrow)$|Detail Description $(\uparrow)$|Complex Rreasoning $(\uparrow)$|All $(\uparrow)$|
> > > > |-----|--------|-----|--------|--------|
> > > > |CLIP|$60.6_{\pm4.0}$|$67.1_{\pm3.7}$|$87.7_{\pm1.9}$|$72.5_{\pm2.0}$|
> > > > |CLIP w/ AFT|$63.2_{\pm3.6}$|$66.3_{\pm1.8}$|$87.8_{\pm2.4}$|$73.3_{\pm1.7}$|
> > > >
> > > > The evaluated scores were comparable w/wo AFT. We would like to clarify that LLaVa-bench is intended to assess the model’s instruction-following capability by evaluating VQA results. However,  AFT aims to improve the model's interpretability, indicated by the visual quality and explainability of the attention maps. Therefore, we do not expect it to significantly enhance performance on LLaVa-bench or similar VQA / caption benchmarks. On the other hand, the results indicate AFT does not sacrifice the quality of the generated response of LLaVa. We recognize the challenge of establishing a comprehensive evaluation framework to quantitatively assess the interpretability of VLMs, which would be an interesting future direction.

---

### Official Review · Reviewer_Hgif · 2024-11-05

**Soundness:** 3
**Presentation:** 3
**Contribution:** 3
**Rating:** 6
**Confidence:** 4

**Summary:**

The paper puts itself in a new area of study — studying interpretability in the context of the vision language models. Whereas the previous works have majorly focused on the interpretation of multimodal interactions, for example, how different text tokens interact with different image regions. However, this paper tries to probe and improve the interpretability of the image encoder of the vision language model (CLIP). They further show that the enhanced interpretability of the vision encoder is transferable - plugging in the same image encoder in a different vision language model (e.g. LLAVA)  provides better attention maps for interpretation. The proposed unsupervised adversarial fine tuning (AFT) method is finally evaluated extensively against various benchmarks and provides mathematical insights into its workings.

**Strengths:**

1. The paper is organized and very well written.
2. The paper reports the evaluation of the proposed AFT approach with good coverage.
3. They show that optimising the upper bound for a sparse set of concepts of vision encoder using L1 regularization is independent of the text embedding and, hence, can be optimized in isolation.
4. They cover the evaluation of saliency map explanations quite well- using transferability, feature importance (ROAR benchmark), attention maps and pointing game.
5. It also provides concept explanation visualization using concept detectors through network dissection.

**Weaknesses:**

1. There are studies that connect adversarial robustness and saliency explanations [2] that attempt to use adversarial training [1] to improve interpretability. Such studies are missing in the related works section in the entirety. Coverage of such studies and then comparing them with the proposes AFT method could significantly enhance the value of the paper and place it in the literature.
2. Because, in real-life scenarios, the features a black box relies on might be non-trivial in image space. For example, both a 'classroom' and a 'movie theatre' have collection of chair, and what differentiates them is the orientation of the furnitures, size of the room and so on. Such concepts can not be highlighted using saliency on the image space. Concept of saliency maps only works in a simple (e.g. object detection) framework.
3. The explanations were initially intended to help the end user (e.g. driver of a car with a deep learning autonomous driving agent) of the application to understand the system better. Such studies determine, irrespective of an explanation being more/less accurate, how much the end user is benefitted from the explanations to perform the enf task [3].


        [1] Xu, Rui, et al. "Scaat: Improving neural network interpretability via saliency constrained adaptive adversarial training." arXiv preprint arXiv:2311.05143 (2023).
        [2] Etmann, Christian, et al. "On the Connection Between Adversarial Robustness and Saliency Map Interpretability." International Conference on Machine Learning. PMLR, 2019.
        [3]  HIVE: Evaluating the Human Interpretability of Visual Explanations (ECCV 2022).

**Questions:**

Refer to the weaknesses.

---

> ### Author Response · Authors · 2024-11-21
> **Authors’ Response to Reviewer Hgif**
>
> We thank Reviewer Hgif for his/her time and thoughtful feedback on our work. Regarding the raised comments and questions,
>
> ---
> >**Related references**
>
> We thank the reviewer for pointing out the related works. We have included the discussion of these related works in the third paragraph of the revised introduction (line 53 and 71). In the revised paragraph, we have discussed the references mentioned by the reviewers and explained in more detail what is different between our proposed AFT and the references’ methods.
>
> To briefly explain the differences between our method and the mentioned works, the mentioned references conduct adversarial training during the training phase of classifier neural networks. Therefore, their proposed methods require the knowledge of classification labels to generate adversarial examples during the training phase. As a result, their methods cannot be directly used for representation learning, because the ground-truth labels are missing for the samples. Moreover, AT for training CLIP from scratch would be computationally difficult, as the CLIP network is trained over huge amounts of data. Finally, we note that the mentioned references have not explored the generalizability and transferability of the effects on interpretation maps. The above points highlight the contributions of our work compared to the existing studies in the literature.
>
> ---
> >**Explaining contextual features of CLIP using saliency maps**
>
> We agree with the reviewer that a uni-modal image-based saliency map may face limitations in explaining the contextual features, e.g. orientation and spatial arrangement, of an input image. However, we have observed that incorporating both text and image modalities within our CLIP-based interpretation framework can address some of these limitations. In the *Appendix A.5* (line 1453) of the revised paper, we present several cases where conceptual features are introduced through the text modality to generate saliency maps. Our numerical results indicate that the CLIP-based saliency maps can capture more complex concepts, such as adjectives related to relative size (e.g., small vs. big) or interactions (e.g., standing vs. holding). For instance, the saliency map highlights the leg region when the prompt is “standing” and the hands when the prompt is “holding.” These findings indicate the potential of the proposed method in interpreting multi-modal models and underscore the importance of advancing research in multi-modal interpretation.
>
> ---
> >**Regarding the end-user benefits**
>
> We thank the reviewer for pointing out the related work on evaluating how saliency maps benefit end-user decision-making. To address the reviewer's comment, we conducted a small-scale user study following [[Kim et al., ECCV 2022](https://link.springer.com/chapter/10.1007/978-3-031-19775-8_17)], focusing on the "distinct" task due to time constraints. In this user study, we selected several cases from the ImageNet validation set and their Grad-CAM maps for the top predicted classes. Users were shown the input image and explanations in random order and asked to select the correct class without seeing the class names. Details are in *Appendix A.4 User Study* (line 1336). We collected responses from 20 users and summarized the results in the following table, reporting the mean accuracy and standard deviation of participants’ performance. The “correct” row refers to cases correctly predicted by the network, while “incorrect” refers to those that were not.
>
> |models|CLIP|CLIP w/ AFT|
> |-----|--------|--------|
> |correct|$73.00_{\pm18.19}$|$74.00_{\pm18.00}$|
> |incorrect|$40.00_{\pm20.00}$|$65.00_{\pm15.33}$|
>
> Our findings indicate that saliency maps aid users in decision-making, with performance metrics well above random guessing (25%). For correctly predicted cases, enhancing saliency map quality had limited impact on user performance, likely because humans can mentally filter out noise. However, for incorrectly predicted cases, higher-quality saliency maps notably improved user decisions. Even when the model's prediction was wrong, the saliency maps guided by the true class name were visually useful and semantically aligned, making users more likely to choose the true class. Conversely, noisy saliency maps led users to ignore relevant information. Additionally, users tended to select the sparsest saliency maps as corresponding to the true class, supporting our aim to enforce sparsity in saliency maps.
>
> A detailed discussion of this user study is provided in Appendix A.4. Although the study was small-scale due to time constraints, it reveals the potential of AFT to enhance end-user decision-making, an important direction for future research.
>
> ---
> **Reference**
>
> - Sunnie S. Y. Kim, Nicole Meister, Vikram V. Ramaswamy, Ruth Fong, Olga Russakovsky.  "HIVE: Evaluating the human interpretability of visual explanations." ECCV, 2022.

---

> > ### Comment · Reviewer_Hgif · 2024-11-27
> >
> > I thank the authors for their extensive rebuttal. I am satisfied with the responses and additional experiments conducted.
> >
> > In case of concerns regarding inability of saliency maps to shed light on the contextual features, the experiments and explanations provided in A.5 discussion are very interesting. However, I am still not sure about the future scope of saliency maps for multimodal interpretability. I feel multimodal interactions are too contextual and complex to be captured by saliency maps.
> >  Even for simple image classification, suppose a deep network classifies an image of a ship sailing in the sea as 'ship'. Would you call the model wrong if it focuses on the ship but also uses the water surrounding as helping context? Similarly, a model may focus on sky apart from an aeroplane to classify an image as 'aeroplane'.
> > As you can see, even in simple object classification setting, the correctness of saliency maps is vague and confusing. Moving to spahisticated tasks like multimodality, it becomes even more critical making the judgement of saliency maps difficult.
> >
> > To summarise, I thank the authors for their effort, additional comments and experiments. And probably, they are correct and rightly carried out. My only concern that continues to remain is, how much salient maps approach can impact in the current day complex problems and features.
> > Thank you again for your great work, and I will keep my score unchanged.

---

> > > ### Author Response · Authors · 2024-11-29
> > > **Thank you for the reply**
> > >
> > > We thank Reviewer Hgif for his/her thoughtful feedback on our response and are glad to hear that it addressed most of the reviewer’s comments. Regarding the reviewer’s remaining comment, we understand the examples mentioned by Reviewer Hgif which indicate cases where saliency maps might not be the optimal approach for interpreting neural networks.
> > >
> > > On the other hand, we would like to emphasize the other use cases where saliency maps are useful for discovering patterns and uncovering the underlying mechanisms of complex systems. In scenarios where sufficient annotated data are available, saliency maps derived from a trained neural network can offer valuable insights and aid scientists in exploring and analyzing these phenomena more effectively.
> > >
> > > Please note that we are not suggesting that scientists should blindly trust saliency maps. Instead, these maps can serve as useful tools for generating hypotheses that can be further investigated and validated. In this way, saliency maps can contribute to a deeper understanding and explanation of phenomena of interest. For example, we show in Fig.10 (line 1102) how the saliency maps help identify suspicious regions of dermoscopy images and X-ray images for further investigation.

---

### Author Response · Authors · 2024-11-21

We would like to thank the reviewers for their time and effort in evaluating our paper and for their valuable suggestions. We are pleased to hear that the reviewers find our work satisfactory for its originality (Reviewers Hgif, jkum, cNDm), presentation (Reviewers Hgif, jkum, Wa4H, cNDm), theoretical analysis (Reviewers Hgif, jkum, cNDm, pAYu), and numerical experiments (Reviewers Hgif, jkum, Wa4H, cNDm, pAYu).

Under each review textbox, we have provided our response to the comments and questions in the review. In our responses, we attempt to clarify the points that were unclear in the original manuscript and provide further analysis on aspects of our method that were pointed out by the reviewers. We have revised the submitted draft in accordance with the reviewers' feedback, and the additions have been highlighted in blue color. We will be happy to discuss any remaining questions of the reviewers in the discussion period.

---

### Meta-Review · Area_Chair_V5PB · 2024-12-20

**Metareview:**

This submission received uniformly positive initial scores from all reviewers, who appreciated the clarity and organization of the paper. However, some concerns were raised regarding the bibliography and the adequacy of quantitative experimental support.

Following the authors' rebuttal, many reviewers acknowledged the effort made to address these concerns and subsequently raised their scores. By the end of the review process, all reviewers expressed a positive opinion of the paper.

After careful consideration, the Area Chair panel has decided to accept this paper.

**Additional Comments On Reviewer Discussion:**

Post rebuttal three reviewers raised their scores and two reviewers had concerns regarding the proposed approach's ability to perform better for VLMs and the use of saliency maps and retained their scores.

---

### Decision · Program_Chairs · 2025-01-22

Accept (Poster)